# A scalable and cGMP-compatible autologous organotypic cell therapy for Dystrophic Epidermolysis Bullosa

Gernot Neumayer[1,2,14], Jessica L. Torkelson [3,4,14], Shengdi Li [5,14], Kelly McCarthy[3,4], Hanson H. Zhen[3,4], Madhuri Vangipuram [1,2], Marius M. Mader [1,2], Gulilat Gebeyehu[6], Taysir M. Jaouni[6], Joanna Jacków-Malinowska [7,8], Avina Rami [7], Corey Hansen[7], Zongyou Guo[7], Sadhana Gaddam[3], Keri M. Tate[4], Alberto Pappalardo [7], Lingjie Li[3], Grace M. Chow[3], Kevin R. Roy [9,10], Thuylinh Michelle Nguyen[9,10], Koji Tanabe[11], Patrick S. McGrath [12], Amber Cramer[3,4], Anna Bruckner[12], Ganna Bilousova[12], Dennis Roop [12], Jean Y. Tang[3,4], Angela Christiano[7], Lars M. Steinmetz [5,9,10], Marius Wernig [1,2,13,15] ✉ & Anthony E. Oro [3,4,15]

We present Dystrophic Epidermolysis Bullosa Cell Therapy (DEBCT), a scalable platform producing autologous organotypic iPS cell-derived induced skin composite (iSC) grafts for definitive treatment. Clinical-grade manufacturing integrates CRISPR-mediated genetic correction with reprogramming into one step, accelerating derivation of *COL7A1*-edited iPS cells from patients. Differentiation into epidermal, dermal and melanocyte progenitors is followed by CD49f-enrichment, minimizing maturation heterogeneity. Mouse xenografting of iSCs from four patients with different mutations demonstrates disease modifying activity at 1 month. Next-generation sequencing, biodistribution and tumorigenicity assays establish a favorable safety profile at 1-9 months. Single cell transcriptomics reveals that iSCs are composed of the major skin cell lineages and include prominent holoclone stem cell-like signatures of keratinocytes, and the recently described Gibbin-dependent signature of fibroblasts. The latter correlates with enhanced graftability of iSCs. In conclusion, DEBCT overcomes manufacturing and safety roadblocks and establishes a reproducible, safe, and cGMP-compatible therapeutic approach to heal lesions of DEB patients.

Over the past decade, advances in the field of stem cell biology and regenerative medicine have enabled the prospect of genetically corrected autologous tissue replacement for previously untreatable conditions. Past clinical successes have mainly used viral gene transfer into somatic tissue, such as the bone marrow stem cells, illustrating that genetic correction of stem cells that are capable of tissue regeneration provides long-term disease-modifying activity[1,2]. Somatic cell reprogramming allows the generation of patient-derived, and thus autologous induced pluripotent stem (iPS) cells that can be genetically manipulated. iPS cells can be differentiated into not only individual cell types but also organotypic cultures or organoids containing multiple key cell types that compose a homeostatic tissue. However, such complex, multi-lineage manufacturing methods have not yet been developed at scale for clinical evaluation[3–5]. The iPS cell-based approach provides a solution for the two main limitations of current somatic cell and gene therapy strategies: (i) iPS cells can be grown to

virtually unlimited numbers, providing a solid foundation for tissue organoid production-scalability and (ii) defined gene editing recreates a wild-type allele while avoiding retroviral insertional mutagenesis. Thus, the combination of cell reprogramming, genomic correction of pathogenic mutations, and composite cell transplantation has the potential to eradicate the impacts of disease-causing mutations in afflicted tissues[5–7].

While appealing in early studies, translation of corrected autologous iPS cell-derived products from proof-of-concept towards realistic clinical manufacturing has been met with a panoply of technical and regulatory roadblocks. Hurdles include developing a robust and reproducible manufacturing method that overcomes critical bottlenecks, including iPS cell generation, validated genetic correction, and safe and effective differentiation into desired tissues. A second set of hurdles includes sequential cell manipulation that results in protracted and labor-intensive manufacturing, increasing batch variability and compromising genomic integrity[8]. Moreover, open questions regarding regulatory concerns, including the determination of the safety risk of a pluripotent cell-derived product, avoidance of animal-derived products, and a risk assessment of genetic mutations introduced during cell culture and genetic engineering, have hampered widespread adoption of this cell/tissue therapeutic platform[9,10].

The skin-blistering disorder Dystrophic Epidermolysis Bullosa (DEB) maps to mutations in the *COL7A1* gene and results in extreme skin fragility due to collagen VII (C7) loss at the basement membrane zone (BMZ)[11–13]. Without curative treatment, the only option remains palliative wound care. Painful chronic wounds severely impact quality of life, and the chronic inflammatory milieu of constantly de- and regenerating skin wounds invariably results in the formation of an aggressive form of squamous cell carcinoma to which most patients ultimately succumb[11–13]. Disease severity and the lack of treatment options motivate the development of scalable and safe manufacturing options for autologous tissue replacement technologies[14–16]. Increasing efforts towards cell and gene therapies to treat Junctional EB have shown great promise, with the basal layer of the skin being self-renewing. Furthermore, correcting the basal keratinocytes that contain holoclones with long-term stem cell activity has been demonstrated to confer remarkable disease-modifying potential[14,17,18]. In addition, a recent Phase I/IIA trial demonstrated that autologous grafts of expanded somatic RDEB keratinocytes transduced with a retrovirally delivered *COL7A1* cDNA has highly efficient wound healing capability[19]. While these groundbreaking clinical trials showed disease-modifying activity, the approaches have important limitations, including difficulty to reliably expand somatic RDEB keratinocytes and the safety concern of insertional mutagenesis by retroviral gene transfer[20,21]. The development of clinically scalable iPS cell-derived skin replacement that overcomes current manufacturing challenges would represent a major advance for many genetic diseases, including DEB.

Here, we realize the advantages of an iPS cell-based multi-lineage differentiation approach to generate organotypic skin composite grafts via next-generation genetic and cellular engineering. Solving critical bottlenecks, we refine a practical and simplified current Good Manufacturing Practice (cGMP)-compatible protocol for the generation of genetically corrected autologous organotypic skin grafts that include keratinocytes, dermal fibroblasts, and melanocytes for the long-term healing of DEB patient wounds.

## Results

### Optimization of CRISPR/CAS9-mediated targeting of the *COL7A1* locus

While previous work[8,22,23] demonstrated the possibilities of ex vivo autologous iPS cell-based gene therapy for the treatment of Recessive Dystrophic Epidermolysis Bullosa (RDEB), several hurdles preventing clinical translation remained. These include (1) relatively inefficient iPS cell derivation and genetic correction, and (2) lack of defined and

efficient protocols for differentiation of edited iPS cells into multi-lineage induced skin composites (iSCs). Consequently, the previous protocols took many months to complete and involved multiple clonal steps, greatly increasing complexity and procedural variabilities, thereby complicating the development of Standard Operating Procedures (SOPs) while increasing the rate of culture-induced mutations[8]. To overcome these limitations, we evolved a next-generation, scalable, non-integrating, xeno-free, and cGMP-compatible platform that produces an epidermal–dermal–melanocyte containing organotypic iSC product for long-term patient wound healing.

We first designed a cGMP-compatible method that allows derivation of *COL7A1*-corrected iPS cells from primary patient fibroblasts in 4 weeks under optimized conditions. This SOP integrates iPS cell reprogramming and gene correction into a single manufacturing step (Fig. 1a), reducing culture time and associated mutational burden, and clonal bottlenecks. In this design, primary patient fibroblasts from a dermal punch biopsy are transiently transfected with (i) CAS9-sgRNA containing ribonucleoproteins (RNPs), (ii) single-stranded oligodeoxynucleotides (ssODNs) that encode the desired genomic correction, and (iii) reprogramming factors-encoding mRNAs that induce iPS cells. iPS cell colonies, emerging 11–14 days (4 patient lines tested; see below) after the initial transfection with reprogramming factors, are then isolated and screened via droplet digital (dd) polymerase chain reaction (PCR) employing probes specific for the properly corrected *COL7A1* locus. Ensuing quality controls validate *COL7A1* edits, cellular identity, and genomic/chromosomal stability.

To test the applicability of this approach, we assessed the potential for gene targeting of the *COL7A1* locus in patient-derived dermal fibroblasts from three individuals carrying the so-called "Colorado" mutation, i.e. *COL7A1* c.7485+5 G > A (Fig. 1b)[24]. Patients CO1 and CO2 carry homozygous, and patient DEB125 carries a compound heterozygous Colorado mutation (*COL7A1* c.6527dupC is the other pathogenicity of DEB125). Initially, we tested all possible sgRNAs mediating CAS9 cutting of the Colorado allele and ssODNs of various lengths encoding for either the (+) or (−) strand of DNA (Fig. 1b). While the 6 possible sgRNAs specific to the Colorado mutation exhibited favorable in silico predicted specificity and activity scores (Supplementary Fig. 1a), Tracking of Indels by Decomposition (TIDE) analysis in patient fibroblasts transfected with RNPs showed variable efficiencies (Fig. 1c and Supplementary Fig. 1b, c). Focusing on the two most efficient sgRNAs (i.e., sgRNA C2 and C4), we analyzed their specificity for the Colorado allele (Fig. 1d). TIDE analysis of wild-type (wt), heterozygous DEB125, and homozygous CO2 fibroblasts transfected with RNPs revealed that sgRNA C4, with a protospacer adjacent motif (PAM) closer to the Colorado mutation, is more specific for the disease allele.

Next, we optimized sgRNA C2 and C4 RNP-mediated repair of mutant *COL7A1* using ssODNs encoding for the wild-type (wt) *COL7A1* sequence and 4 silent mutations used for detection of editing events via specific ddPCR probes (Fig. 1b). By comparison with a bi-allelic reference locus, ddPCR allowed quantification of the edited *COL7A1* alleles. This approach indicated that sgRNA C2 mediated 2–2.5× more repair of *COL7A1* than sgRNA C4 (Fig. 1e). The ssODN length (up to 200 nt) positively correlated with editing efficiencies (Supplementary Fig. 5b, c). Surprisingly, also the orientation of the employed ssODNs influenced editing efficiencies, with the (+) strand-encoding sequence yielding 2–2.5× higher efficiencies than the (−) ssODN in both homozygous (Fig. 1e) and heterozygous patients (Supplementary Fig. 1d). We confirmed these ddPCR results by cloning the target locus of bulk-edited fibroblasts into plasmids, followed by analysis of individual, cloned alleles via PCR primers specific for the edited locus (Fig. 1f and Supplementary Fig. 1f–h). Analysis of 77 individual target alleles each, from cells treated with all possible combinations of sgRNA C2 or C4 and ssODN (+) or (−), validated the ddPCR results. Sanger sequencing revealed that both, sgRNA C2 and sgRNA C4 can mediate *COL7A1*

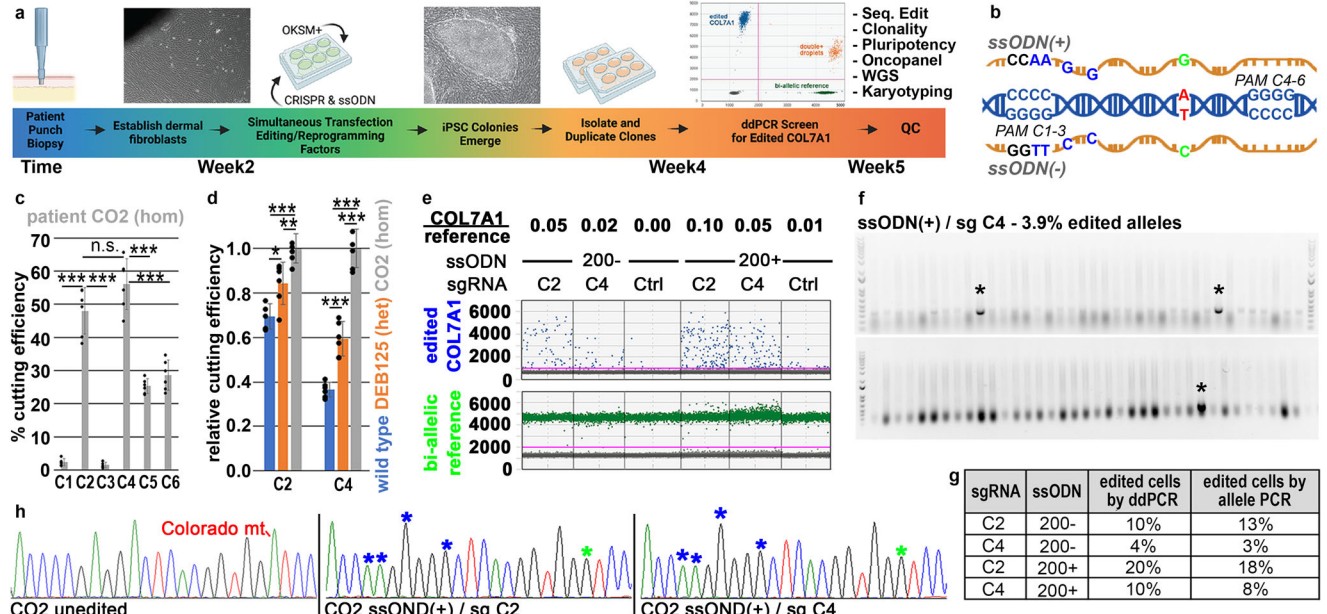

**Fig. 1 | Optimized editing of the *COL7A1* Colorado mutation (7485 + 5 G > A).**
**a** Overview of single-step editing/reprogramming. **b** Overview of the Colorado mutation (red) and ssODNs (4 silent mutations (blue); wild-type sequence (green)) used for gene editing. sgRNAs (C1–C6) engage 6 possible PAMs that mediate cutting via CRISPR/CAS9. **c** Absolute CRISPR/CAS9 cutting efficiencies in CO2 patient fibroblasts as mediated by sgRNA C1–C6 (*n* = 6 biological replicates; mean and SD are shown with individual data points overlayed as scatter plot). **d** Relative CRISPR/CAS9 cutting efficiencies of the homozygous (CO2), heterozygous (DEB125) Colorado or wild-type allele as mediated by sgRNA C2 or C4 in indicated patient fibroblasts (*n* = 6 biological replicates, except wild-type sgRNA C2 *n* = 5; mean and SD are shown with individual data points overlayed as scatter plot). **c, d** Statistical significances calculated via two-tailed homoscedastic *t* tests (**P* < 0.05, ***P* < 0.01, ****P* < 0.001, n.s. not significant; *P* values: **c** C1/C2 2.3E-08, C2/C3 1.8E-08, C1/C3 1.1E-01, C4/C5 2.6E-06, C5/C6 1.4E-01, C4/C6 1.9E-05, C2/C4 8.7E-02; **d** C2: wt/125 1.3E-02, wt/CO2 1.9E-05, 125/CO2 7.3E-03; C4: wt/125 5.5E-05, wt/CO2 1.2E-08, 125/CO2 6.2E-06). **e** *COL7A1* editing efficiencies measured by ddPCR in CO2 primary patient

fibroblasts after transfection with ssODNs and sgRNA/CAS9-containing RNPs as indicated. A bi-allelic locus (green) is used as a reference for calculating *COL7A1* editing (blue) efficiencies. Ctrls omitted sgRNAs. **f** Agarose gels visualizing 77 *E. coli* colony PCRs detecting edited, Topo-cloned *COL7A1* alleles from cells treated as in (**e**) with ssODN/sgRNA combinations as indicated (see Supplementary Fig. 1 for remaining ssODN/sgRNA combinations). A primer specific for silent mutations (**b**) only yields PCR products of edited alleles (asterisks). DNA size references were run in most left (top/bottom) and right (top) lanes (100–2000 bp ranges shown). **g** Summary of *COL7A1* editing efficiencies achieved with different ssODN/sgRNA combinations as measured by ddPCR (**e**) or by Topo cloning of individual alleles (**f**). Mono-allelic edits were assumed for calculations (see discussion). **h** Sanger sequencing traces of unedited and edited alleles from (panel **f** and Supplementary Fig.1h). Asterisks indicate integration of intended silent mutations (blue) and repair of the pathogenic mutation (green). Source data are provided as a Source Data file. Panel **a** created with BioRender.com released under a Creative Commons Attribution-NonCommercial-NoDerivs 4.0 International license.

editing as intended when used in combination with (+) ssODNs (Fig. 1g, h and Supplementary Figs. 1f–h and 2f, g). Remarkably, although all the Sanger-sequenced alleles from cells treated with (−) ssODNs exhibited at least partial integration of donor sequences, none of them was correctly edited. We conclude that all parameters involved, i.e., the sgRNA sequence, the length and orientation of ssODNs, and the particular target locus must be tested for optimal efficiency and specificity of editing events.

## Combining iPS cell reprogramming and *COL7A1* correction in one manufacturing step

The optimized *COL7A1* targeting efficiency allowed us to test whether it may be feasible to correct and reprogram cells in a single manufacturing step, which would greatly minimize production time and eliminate multiple clonal selections. We chose to deliver the reprogramming factors via transfection of mRNA since this approach has been shown to be efficient, is compatible with cGMP manufacturing using chemically defined reagents, and represents a transient treatment leaving no genetic scars[23]. In line with the TIDE assay in fibroblasts (Fig. 1d), sgRNA C2-mediated *COL7A1* editing in DEB125 fibroblasts directly followed by mRNA reprogramming resulted in ~15% targeted iPS cell colonies but was not specific for repair of the mutant allele of this heterozygous patient (Supplementary Fig. 2a–e). In addition, next-generation sequencing of PCR amplicons generated from edited primary patient fibroblasts revealed that most alleles with integration of donor sequence had incorporated all (or some) of the

designed silent mutations used for genotyping by ddPCR but had failed to repair the Colorado mutation that is located more distantly to the sgRNA C2-mediated CAS9 cut site (see Fig. 1b and Supplementary Fig. 2f–h). This incomplete integration of donor sequence results in false positives by ddPCR (Supplementary Fig. 2d; i.e. integration of silent 'indicator' mutations but failure to repair the genetic pathogenicity). Similarly, sgRNA C4-edited alleles that had repaired the Colorado mutation proximal to the cut site showed less efficient integration of the designed silent "indicator" mutations, located more distally to the chromosomal break (Supplementary Fig. 2g). These data indicate a drastic drop of donor integration efficiency with increasing distance to the CAS9 cut site. However, we also note locus-specific differences, as in comparison to sgRNA C2, sgRNA C4-edited fibroblasts harbored approximately twice (>1.9×) as many alleles with both, repaired Colorado loci and integration of all four designed silent "indicator" mutations (i.e., complete integration of donor-mediated edits; Supplementary Fig. 2f, g). We therefore focused on sgRNA C4, which is also more specific for the Colorado allele (Fig. 1d).

We determined via a dose range that 5 pmole sgRNA C4-containing RNP per 30k fibroblasts was sufficient for optimal cutting of the Colorado allele (Supplementary Fig. 1e). Next, we measured the *COL7A1* editing efficiencies in fibroblasts of patients CO1 and CO2, which carry a homozygous Colorado mutation, using engineered high-fidelity CAS9s. Reassuringly, ddPCR, using ssODN (+) and sgRNA C4 in complex with CAS9s HiFi or SpiFy showed targeting events in 3–7% of the cells (Fig. 2a, b). These encouraging efficiencies prompted us to

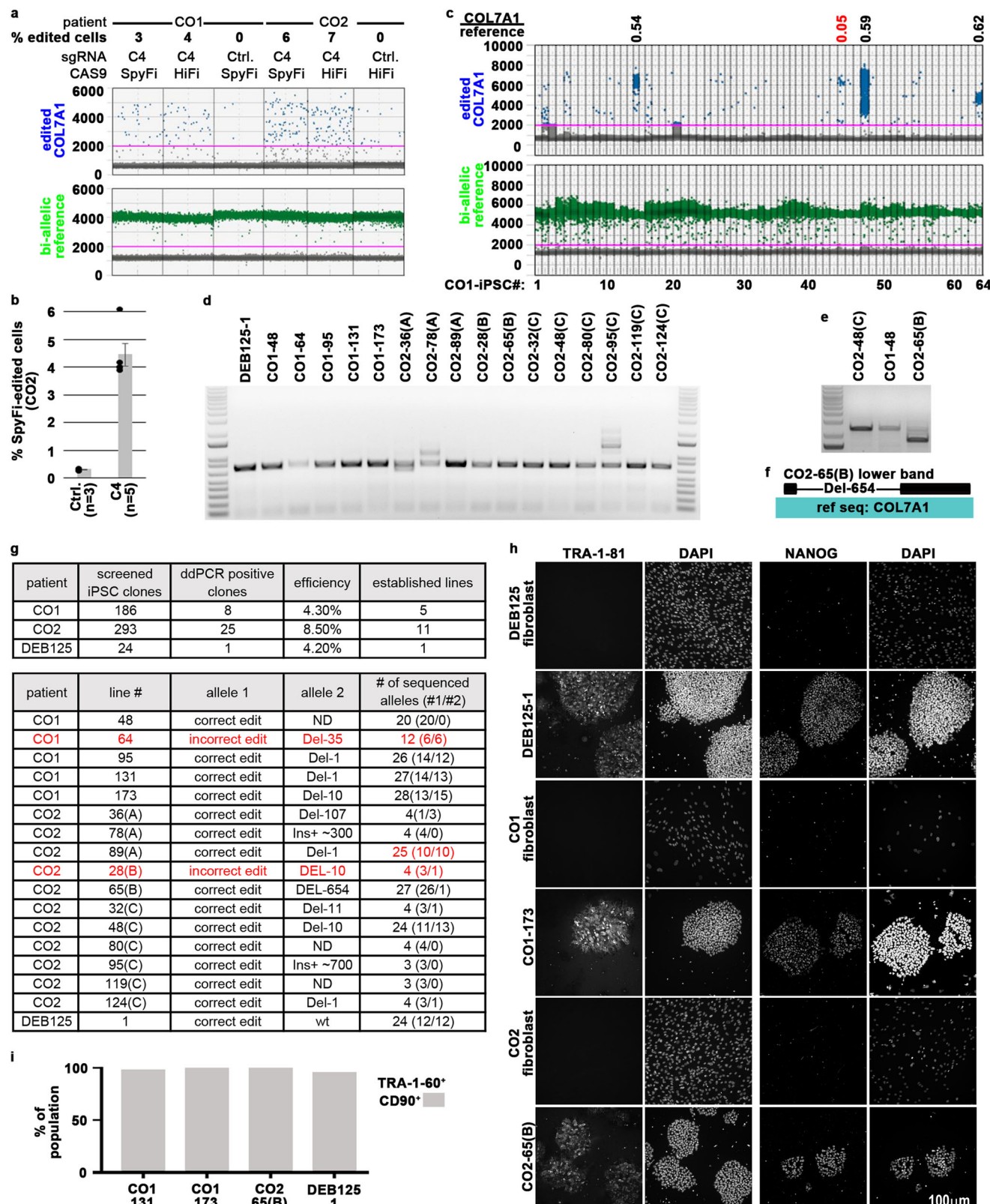

test the induction of reprogramming immediately following *COL7A1* editing via cGMP-compatible SpiFy CAS9. We screened 186, 293, and 24 iPS cell clones derived from patient CO1, CO2, and DEB125 by ddPCR, yielding 8, 25, and 1 candidate lines, respectively (Fig. 2c, g and Supplementary Fig. 4). Analysis of 5, 11, and 1 candidate iPS cell lines from each patient via conventional PCR amplification of the target locus, followed by plasmid cloning and Sanger sequencing of

individual alleles, revealed that all but 2 lines were correctly edited (Fig. 2d–g). As expected from the specificity and efficiency mediated by guide C4 (Fig. 1d), in iPS cell lines from patients homozygous for the Colorado mutation (i.e., CO1 and CO2) the second allele acquired insertion or deletion (InDel) mutations of various sizes in all cases, whereas the second, wt allele in iPS cells derived from the heterozygous patient DEB125 remained unperturbed. Of important note,

**Fig. 2 | Successful single manufacturing step editing/reprogramming of patients CO1, CO2, & DEB125. a** *COL7A1* editing efficiencies measured by ddPCR in CO1 and CO2 patient fibroblasts after transfection with ssODN(+) and RNPs containing sgRNA C4 and high-fidelity CAS9 HiFi or SpyFi as indicated. A bi-allelic locus (green) is used as a reference for calculating *COL7A1* editing (blue) efficiencies, assuming mono-allelic editing events. Ctrls omitted sgRNAs. **b** Reproducible *COL7A1* editing in CO2 patient fibroblasts (as in (**a**)) with SpyFi CAS9 (*n* = biological replicates as indicated; mean and SEM are shown). **c** ddPCR screen of 64 single-step edited/reprogrammed iPS cell lines derived from patient CO1 fibroblasts. Ratios of edited *COL7A1* alleles (blue) and a bi-allelic reference locus (green) are used to identify mono- (0.5 +/−0.19) or bi-allelic (1.0 +/−0.19) editing events (black values; red values below/above cutoff indicate mixed or incorrectly edited clones; see Supplementary Fig. 4). **d**, **e** Agarose gels visualizing PCR amplicons of a 731 bp (**d**) and 2418 bp (**e**) sequence surrounding the edited *COL7A1* locus from single-step edited/reprogrammed iPS cell lines derived from three patients. Note some

samples yield 2 PCR products, indicative of InDels on one of the *COL7A1* alleles. InDels can be substantial (e.g., line CO2-65(B)), so they are only included on bigger (**e**) PCR products. DNA size references were run in most left (**d**, **e**) and right (**d**) lanes; 100–15,000 bp (**d**) or 1500–15,000 bp (**e**) range is shown. **f** Sanger sequencing of the smaller PCR product from line CO2-65(B) from (**e**) reveals a large 654 bp deletion. **g** Summary of single-step editing/reprogramming screens conducted with sgRNA C4/ssODN(+) from three patients as indicated (top). Topo cloning and sanger sequencing of PCR products (**d**–**f**) confirm correct *COL7A1* editing on target alleles in 15 of 17 single-step edited/reprogrammed iPS cell lines. **h** Immunofluorescence microscopy images of iPS cells and parental fibroblasts stained for pluripotency markers TRA-1-81 and NANOG from three patients. DAPI visualized DNA, scale indicated. **i** Summary of flow cytometry analysis of iPS cells from three patients for CD90 and the pluripotency marker TRA-1-60 (see Supplementary Fig. 3d). Source data are provided as a Source Data file.

in some iPS cell clones derived from homozygous Colorado patients the InDels on the second allele were larger than the 731 bp PCR amplicon that we initially chose for analysis (Fig. 2d), wrongly implying bi-allelic *COL7A1* correction. Larger PCR amplicons identified some of these bigger InDels, e.g. a 654 bp deletion in line CO2-65(B), while the nature of others (e.g. line CO1-48) could not be determined (Fig. 2e–g; see "Discussion").

Next, we chose candidate iPS cell lines for deeper genomic characterization. Sanger sequencing of up to 28 individual target loci cloned into plasmids showed an equal distribution between correctly edited and the second allele in all but one sample (i.e. CO2-89(A); Fig. 2g). In iPS cell lines derived from homozygous patients, the second allele that did not incorporate the ssODN always displayed a characteristic InDel, indicating high efficiency of employed RNPs. Other than iPS cell line CO2-89(A), in which 20% of sequences displayed deletions of a different nature, our data is consistent with clonal origin of picked iPS cell lines. In addition, we verified the cellular identity of our iPS cell lines by robust expression of the pluripotency markers TRA-1-81, TRA-1-60, NANOG, *LIN28*, *OCT4*, and *SOX2* as determined by immunofluorescence, flow cytometry, and/or RT-PCR (Fig. 2h, i and Supplementary Figs. 3d and 6a).

Finally, to test applicability to correct other mutations, we followed the same SOPs developed for the Colorado mutation and successfully derived one-step corrected iPS cell lines with similar efficiency from a fourth patient, DEB135, who carries two different pathogenic compound heterozygous mutations, i.e. *COL7A1* c.6781 C > T and c.6262 G > A (Supplementary Fig. 5). Importantly, this demonstrates that upon adjusting the CRISPR/CAS9-mediated editing strategy to the patient-specific mutation our single clonal step iPS cell manufacturing process can be adapted for other therapeutic cell manufacturing procedures.

## Scalable and reproducible differentiation of DEB iPS cells into organotypic skin grafts

Proper mammalian skin development requires the interaction between early surface ectoderm progenitors, regional mesoderm, and neuroectoderm[25–27]. In particular, we recently demonstrated the importance of a subtype of mesoderm that depends upon the chromatin regulator Gibbin for proper epidermal stratification[28]. We therefore sought to develop an advanced multi-lineage cutaneous organoid differentiation method that imitates the interaction and co-dependence of the cell lineages that cooperate in the embryo to form skin, such as that between developing ectoderm and Gibbin-dependent mesoderm (Fig. 3a)[28,29]. We initially used H9 human embryonic stem cells to optimize our methodology. First, embryonic surface ectoderm was induced with retinoic acid/bone morphogenetic protein-4 (RA/BMP-4) for 7 days[28,30]. scRNA-seq analysis of day 7 cultures showed the successful creation of surface ectoderm, mesoderm, and neuroectoderm (Fig. 3a, b and Supplementary Fig. 6f). The second inductive phase used

a defined matrix and media containing the epidermal growth factors insulin, EGF and FGF for an additional 40–45 days, allowing reciprocal epithelial–mesenchymal–neuroectodermal interactions to mature the cultures into a therapeutic organotypic induced skin composite (iSC).

A common hurdle in pluripotent cell differentiation comes from stochastic mechanisms during complex cell culture that lead to variable keratinocyte maturation and the presence of immature K14+, K18+ epithelial cells (Supplementary Fig. 6b, c, f)[31]. To overcome this hurdle and to enrich for mature basal keratinocytes, we used the previously characterized stem cell surface markers ITGA6 /ITGB4[32,33]. We verified that ITGA6 is expressed highly on p63+; K14+; K18- cells, while K18+ cells were ITGA6 low or negative. An ITGA6 magnetic bead-based, automated pro-separator AutoMACS® efficiently enriched for p63+; K14+ cells and removed K18+ cells (Fig. 3c–f and Supplementary Fig. 6c). In contrast to non-enriched populations, ITGA6-enriched cells produced robust stratified epidermis/dermis in liquid/air interface organotypic cultures as demonstrated by involucrin expression and deposition of C7 to the BMZ (Fig. 3c–e). Quantification of patient-derived *COL7A1*-corrected iPS cell lines, iPS cell lines with no known genetic pathogenicity (i.e., WTC-11 and DSP), and H9 ES cells by flow cytometry allowed us to determine the ratio between input iPS cells and ITGA6+ cells after differentiation and enrichment, i.e. the "coupling efficiency" metric (Fig. 3g, h and Supplementary Fig. 6d). This analysis indicated comparable but distinct coupling efficiencies, demonstrating robustness of our methodology but also highlighting line-to-line variability, even between cell lines with no known pathogenic mutations. Notably, we successfully differentiated and AutoMACS®-enriched five independent, genetically corrected DEB patient cells lines derived from individuals with distinct genetic pathogenicities in multiple replicates using cGMP-compatible materials (Fig. 3h).

In anticipation of a future clinical trial, we sought to implement ITGA6 enrichment at a clinical manufacturing scale. We performed five large-scale differentiation runs that improved iSC formation by employing a CliniMACS® Plus cell separator in three different modes, yielding various enrichment and cell viability ratios (Supplementary Fig. 6e). With the high observed coupling efficiencies, only limited cell expansion in vitro would be necessary to generate the needed cell numbers for future clinical trials (Supplementary Fig. 6e–j). Flow cytometry and bulk RNA-seq verified successful enrichment (Supplementary Fig. 6f–j), demonstrating the feasibility of clinical-scale manufacturing.

Lastly, we used single-cell transcriptomics to characterize the final post-enrichment iSCs, which would be equivalent to the DEBCT clinical product substance. Importantly, mesodermal (and melanocytic) cell populations were still present after ITGA6 enrichment, allowing continued signaling between cell types (Fig. 3i, j). We profiled iSCs from H9 cells and five patient iPS cell lines, revealing 8 distinct cell clusters representing surface ectodermal (C1, C2, C4, C7), mesodermal (C3, C5, C8) and melanocyte cell types (C6) (Fig. 3i, j and Supplementary

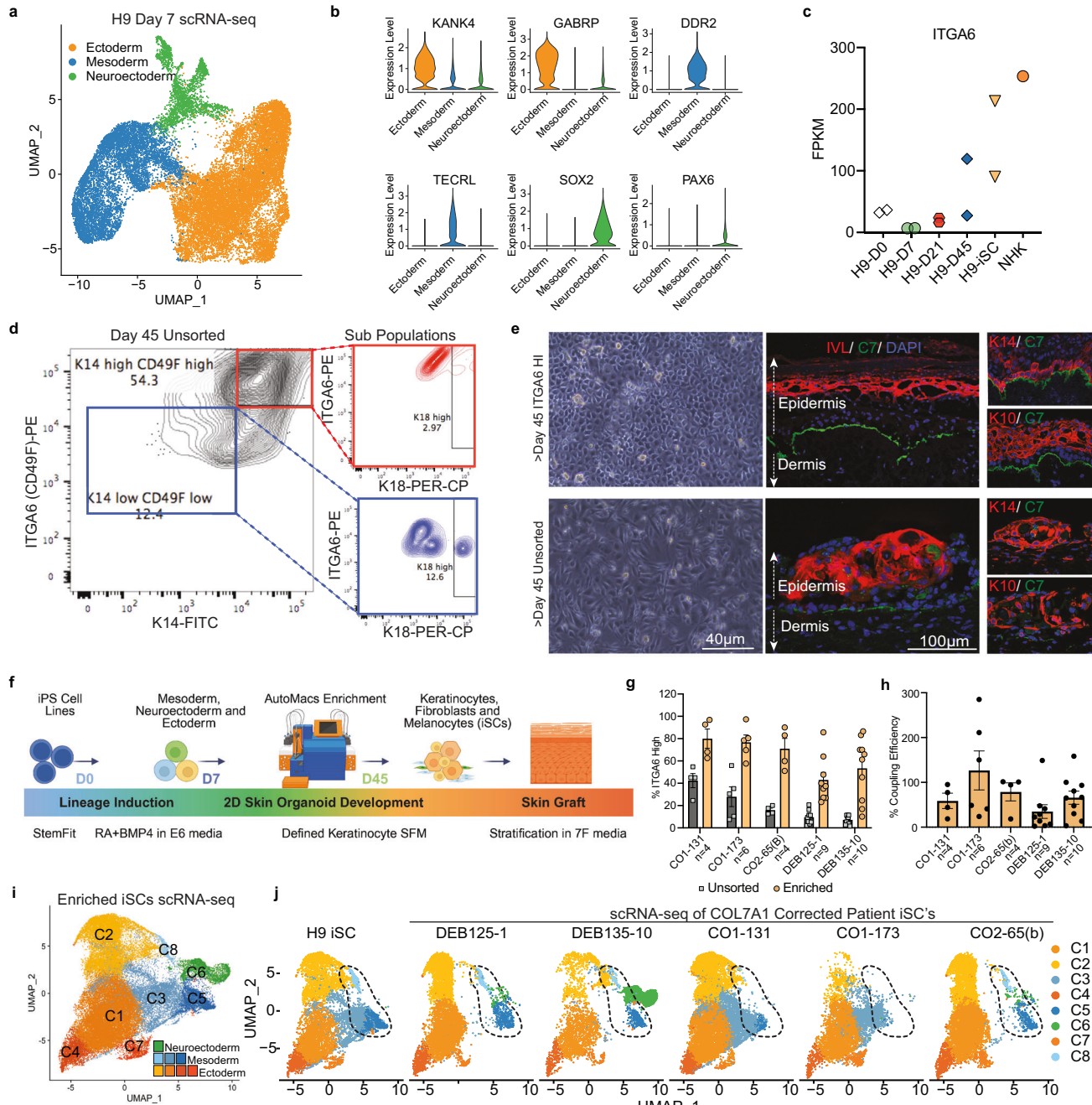

**Fig. 3 | Generation of organotypic skin grafts at clinical scale. a** Uniform manifold approximation projection (UMAP) of integrated scRNA-seq of H9 ES cell differentiation towards iSCs on Day 7 reveals three lineages: ectoderm (orange), mesoderm (blue), and neuroectoderm (green). **b** Violin plots depicting relative expression (RPKM) of representative ectoderm, mesoderm and neuroectoderm genes across scRNA-seq clusters. **c** ITGA6 expression analyzed via RNA-seq in H9 ES cells during iSC differentiation. Time points as indicated (D: day; $n = 2$ biological replicates; NHKs used as positive control $n = 1$). **d** Flow cytometry of day 45 unsorted H9 ES cells. Cells double positive for ITGA6$^{HI}$ PE (y-axis) and K14$^{HI}$ FITC ($x$ axis) are in high gate (red) and lower ITGA6/K14$^{LOW}$ expressing cells are in low gate (blue); expression of K18 (PER-CP; x axis) in subpopulations (right). $n = 1$. **e** Bright-field image (top left) of FACS sorted ITGA6$^{HI}$ H9 ES cell-derived iSCs expanded and used for organotypic stratification (top right). Note normal polarization and stratification; Collagen 7 (green), Involucrin (red), DAPI (blue); additional images, K14, K10 (red). Bright-field image (bottom left) of unsorted H9 ES cell-derived iSCs used for organotypic stratification (bottom right). Note disorganized layering and stratification. $n = 1$, scale indicated. **f** iPS cell to iSC differentiation strategy employing cell enrichment via AutoMACS pro-separator, following a defined cGMP-compatible protocol. **g** Flow cytometry analysis of % ITGA6 positivity measured before and after AutoMACS enrichment. **h** % Coupling efficiency (CE) determined by the equation (5) %CE= live sorted iSCs/iPS cell input*100. **g, h** Data from five independent differentiated patient cell lines ($n$ = indicated; mean, SEM;). **i** UMAPs of integrated scRNA-seq data from five patient- and H9 ES-derived iSCs post-ITGA6 enrichment and in vitro expansion reveal 8 clusters (C1–C8) comprising the DEBCT product. **j** Individual UMAP plots from overlaid scRNA-seq datasets from (**i**) with color scheme as indicated. Four ectoderm (C1, C2, C4, C7), 3 mesoderm (C3, C8, C5), and 1 melanocyte/neuroectoderm-derived (C6) clusters were identified. C5/C8 clusters indicated by dotted outline were present at variable quantities (see text, Fig. 5 and Supplementary Fig. 7). Source data are provided as a Source Data file. Panel **f** created with BioRender.com released under a Creative Commons Attribution-NonCommercial-NoDerivs 4.0 International license.

Fig. 7a, b)[28,34–38]. C1 most closely resembles basal keratinocytes (high K14, low K18, low cell cycle markers), while C2 resembles long-term proliferative "holoclone" keratinocyte stem cells (high CDK1 and TOP2A; Supplementary Fig. 7a, c, d)[39,40]. Of note, the fraction of C2 holoclone-like cells in DEBCT is much higher than in cultured somatic skin preparations used to isolate holoclones, reinforcing an advantage of iPS cell-based manufacturing (Supplementary Fig. 7d). Cells in C4 and C7 express initiated signatures of the early epidermal stratification phase. C5 and C8 cells closely resemble Thy1/CD90+ Gibbin-dependent mature dermal fibroblasts that are required for epidermal stratification (Supplementary Fig. 7a–e and Supplementary Data 1)[28], and the C3 cluster contains cells of more immature dermal/pre-vascular characteristics[34]. Intriguingly, the six cell lines produced different ratios of these induced cell clusters, allowing us to correlate the implications of their presence/absence with DEBCT-product performance (see below; Figs. 3j and 5c–f).

## Genomic stability of *COL7A1*-corrected iPS cells and iSCs

A critical safety aspect of cell expansion is genomic and chromosomal stability, as we estimate that $3 \times 10^7$ undifferentiated iPS cells are needed to generate one clinical iSC application of a $6 \times 8$ cm sheet graft in a Phase I/IIa trial (Supplementary Fig. 6h–j)[16]. Unlike previously tested media[8], a more recently developed chemically defined media on plates coated with the E8 fragment of Laminin-511[41] allowed expansion of karyotypically normal iPS cell lines derived from four individuals with three sgRNAs to at least $3 \times 10^7$ cells in 10 of 11 instances (Fig. 4a and Supplementary Fig. 8a). As part of our product safety methodology, we performed whole-genome sequencing (WGS, 40x coverage) of four single-step manufactured, *COL7A1*-corrected iPS cell lines from three patients carrying the Colorado mutation, their parental fibroblasts, and differentiated ITGA6-enriched iSCs. WGS data confirmed all CAS9-mediated *COL7A1* edits (Fig. 2g). To identify single nucleotide variants (SNVs) and InDels with high confidence, we used the agreement between three variant callers and filtered out all SNVs/InDels previously annotated in SNV/InDel databases (see "Methods"). This strategy yielded an average of 70,386 +/−570 (SEM) SNVs and 32,465 +/−1506 (SEM) InDels in each of the 11 samples. To find SNVs and InDels that are present specifically in either iPS cells and/or iSCs and thus, may be induced and/or selected for by our manufacturing process, we employed two alternative and complementary strategies: (i) *k*-means clustering by allele frequency (AF) and (ii) AF/odds ratio cutoff filtering. The *k*-means clustering at high resolution (feature space *k* = 9) showed that most SNVs/InDels exhibit similar AFs among all three cell types, suggesting that these variants are heterozygous or homozygous pre-existing germline variants (Fig. 4b and Supplementary Fig. 8b). Only one cluster (cluster 7 for DEB125-1, Fig. 4b) contained iPS cell/iSC-specific variants and no cluster was identified with variants unique to either iPS cells or iSCs. The same pattern was observed in all four experiments. Importantly, there were only three iPS cell/iSC-specific variants found in more than one patient (Supplementary Fig. 8c). Of these three variants, one shared between DEB125 and CO2 maps to an intron of *Magi2* and is already present in the respective parental fibroblasts at AF = 0.1–0.2. Similarly, the two variants shared between lines CO2-65(B) and CO1-173, mapping to an intron of a ncRNA (i.e. LOC100507053) and an intergenic region, are also found in parental fibroblasts at AF = 0.2–0.3, and AF = 0.1–0.2, respectively. Thus, all three variants, themselves of unknown significance, were in fact already pre-existing in parental fibroblasts at lower AFs and are not introduced de novo by our manufacturing process.

Reassuringly, our second strategy based on AF/odds ratio cutoff filtering identified >89% of the variants called by *k*-means clustering as shared iPS cell/iSC-specific (Supplementary Fig. 8d). Unlike *k*-means clustering, AF/odds ratio cutoff filtering identified variants unique to iPS cells or iSCs only, albeit in much smaller numbers than shared iPS cell/iSC-specific variants (Fig. 4c). Of note, almost all iPS cell-specific

variants were present in other cell types with an AF > 0, suggesting that these variants were not introduced de novo (Supplementary Fig. 8e). In contrast, shared iPS cell/iSC- and iSC-specific variants were infrequent in other cell types suggesting some of them are de novo or amplified from a rare (AF < 5%) pre-existing variant (Supplementary Fig. 8e). There was no overlap of these variants between patients (Fig. 4c). Most cell-type-specific variants map to intronic or intergenic regions. Gene ontology (G.O.) term analysis of all variants located at loci with predicted functional properties (i.e., exons, splice sites, UTRs, and promotor/enhancer elements) did not yield any significant enrichment (Fig. 4c).

To investigate potential guide-dependent CAS9 off-target mutations, we searched a window of 25 bases around all shared iPS cell/iSC variants for PAM-like NRG motifs that are combined with sequence similarities to the used sgRNA C4[42]. We did not observe any NRG motif adjacent to a 20-mer containing less than 6 mismatches compared to the sequence of sgRNA C4 (Fig. 4d). The cutting efficiency of CAS9 misguided by sgRNAs containing more than 3 mismatches has been reported to be exceedingly low[43,44]. Accordingly, TIDE analysis of bulk-edited fibroblasts transfected with sgRNA C4/CAS9 RNPs did not detect any significant InDel formation at any in silico predicted exonic or intronic off-target, whereas the *COL7A1* on the target site was cut with over 87% efficiency (Fig. 4e, g). As TIDE can detect a maximum InDel size of 50 bp, we also visualized InDels via plotting the normalized coverage of WGS reads obtained from unperturbed fibroblasts, thereof derived iPS cells and iSCs around all in silico predicted exonic, intronic, and intergenic off-targets and the *COL7A1* on-target. All previously identified CAS9-mediated deletions on the non-repaired *COL7A1* allele displayed the expected decreased read coverage (Fig. 4f, compare with Fig. 2g). No other InDels in proximity of potential off-target cut sites were observed when searching a 1 kb or 1 Mb window (Fig. 4g).

The functional consequences of detected variants are hard to predict. Their random nature and frequent pre-existence in the heterogeneous patient fibroblasts suggest that pathogenic effects caused by our manufacturing process are unlikely to materialize. To exclude, however, the de novo introduction or clonal expansion of variants in potentially cancer-promoting genes we performed the Clinical Laboratory Improvement Amendments (CLIA)-accredited, high-coverage sequencing Stanford Actionable Mutation Panel for Solid Tumors (STAMP)[45]. iPS cell lines DEB125-1, DEB135-10(B), DEB135-24(B), CO1-131 and CO1-173 displayed no STAMP-hits other than germline variants already present in unperturbed parental fibroblasts (Fig. 4h and Supplementary Fig. 5h). iPS cell lines CO2-65(B) and CO2-48(C) and their iSC progeny exhibited a heterozygous mutation in the androgen receptor (AR) that STAMP did not detect in fibroblasts (Fig. 4h). Targeted ddPCR analysis of this mutation in parental fibroblasts revealed however an AF of 3% of this variant (which is below the 5% detection limit of STAMP), suggesting clonal expansion of a rare somatic mutation or even a pre-cancerous lesion of this patient. Accordingly, many other iPS cell lines derived from patient CO2 by the same production run did not harbor this AR-mt (Supplementary Fig. 8f, g). These results highlight the merit of the STAMP oncopanel to exclude rare cell products with potentially pathogenic mutations.

## In vivo efficacy and favorable safety profile of patient-derived *COL7A1*-corrected organotypic skin grafts

To test the functionality of the corrected organoid iSCs, we transplanted ITGA6-enriched iSC cultures on the back of immunocompromised nude mice. With 3–12 manufacturing runs per line, all five *COL7A1*-corrected patient iPS cell lines produced viable grafts and formed human skin in vivo (Fig. 5a, b). Graft success was determined by formation of stratified epidermis consisting of K14+ basal cells and K10/Involucrin+ upper stratified layers, and detection of human-specific C7 in the BMZ (Fig. 5b). Successful grafts were stable for at

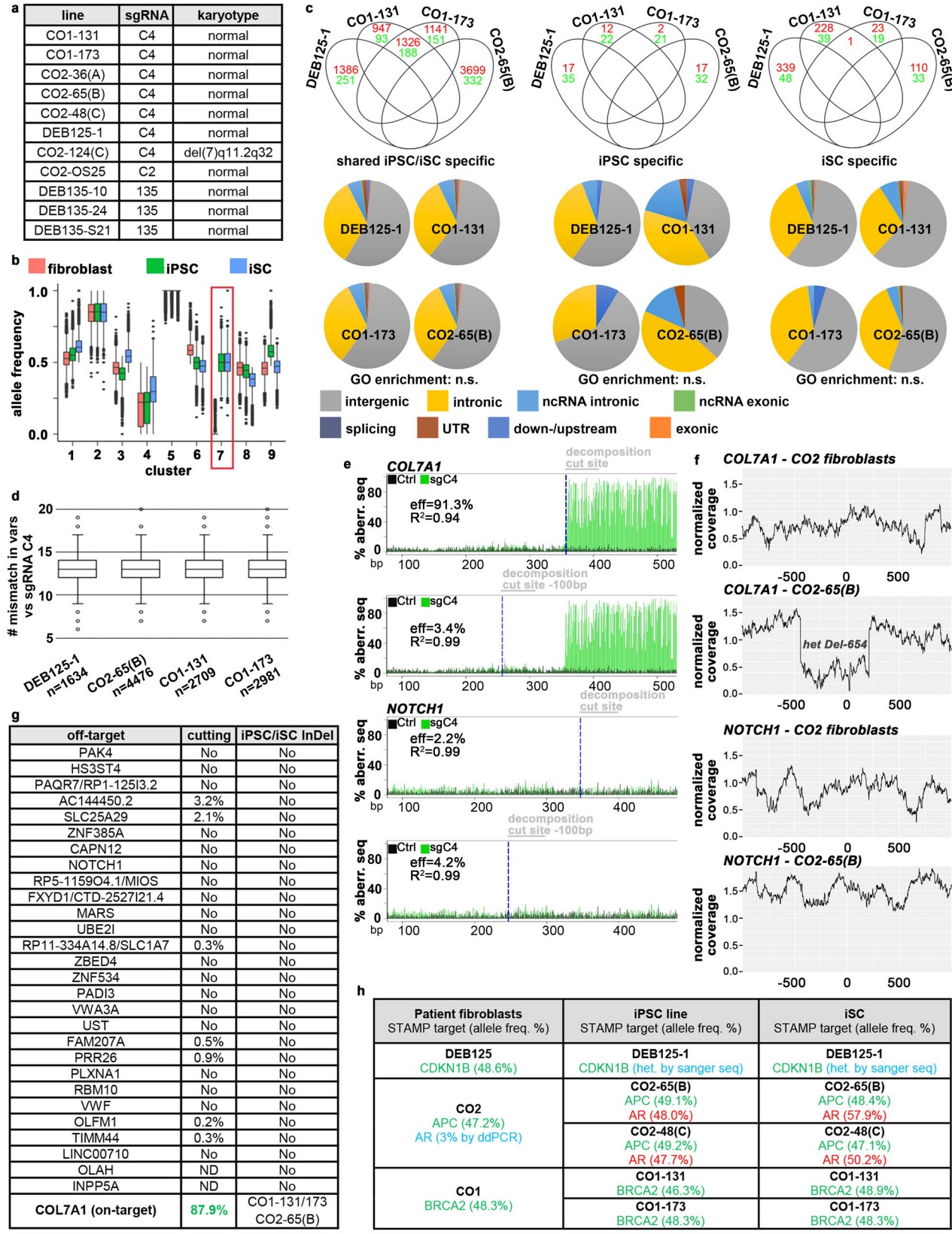

least 1 month after transplantation and comparable to primary human keratinocyte grafts[8,46,47], thereby providing critical biologically and clinically relevant quality attributes.

While our manufacturing led to graftable iSCs from each line, we observed a line-dependent range of grafting efficacy (Fig. 5a). The different distributions of cell identities among the five lines

characterized (Fig. 3i, j) enabled us to correlate the four epidermal, three dermal, and one melanocyte cell groups of iSCs with observed engraftment success. One prominent variable was the melanocyte cluster but that did not correlate with graftability (Figs. 5c and 3i, j). However, the lines with lowest grafting efficiency (CO1-173, CO1-131) distinctly lacked the two Gibbin-dependent fibroblast populations C5/

**Fig. 4 | Genomic and chromosomal stability. a** Normal karyotypes in 10 of 11 iPS cell lines from four patients. **b** *k*-means clustering of all variants (*n* = 111,741) found by whole-genome sequencing (WGS) in fibroblasts, iPS cells and iSCs from patient DEB125 (see Supplementary Data 2). Red frame highlights a sub-set with differential allele frequencies (AFs) in iPS cells/iSCs compared to fibroblasts. Boxes: inter-quartile ranges (25–75%), center lines: medians, whiskers: 1.5 times the interquartile range; outliers: circles. **c** Cutoff/odds ratio filtering (see Methods) identifies variants from WGS (red: SNPs; green: InDels) specifically found in cell types from indicated patient lines. Grouping in Venn diagrams indicates the absence of positive variant selection. The majority of cell-type-specific variants is found in intergenic or intronic sequences (pie charts). No gene ontology (GO) term enrichment detected. **d** Aligning all shared iPS cell/iSC-specific variants with the used seed sequence of sgRNA C4 within a 25 bp search window that must contain a NRG PAM-motif does not identify any homologies. The outlier (circles) with the highest similarity exhibits six mismatches. Boxes: interquartile ranges (25%–75%), center

lines: medians, whiskers: 1.5 times the interquartile range. **e** TIDE analysis of sgRNA C4-mediated CAS9 cutting in CO2 fibroblasts at the *COL7A1* on-target (top) and the in silico predicted off-target *NOTCH1* (bottom). Measurements taken upstream (Ctrl) and downstream of cut sites. **f** Plots of normalized WGS coverage 1000 bp up-/downstream of *COL7A1* (top) and *NOTCH1* (bottom) from fibroblasts/iPS cells. Note the sharp drop of coverage at the *COL7A1* locus in iPS cells due to a heterozygous 654 bp deletion (Fig. 2d–g). **g** Summary of all in silico predicted exonic and intronic off-targets as in (**e**, **f**). For TIDE (middle), controls were subtracted from cut sites. Heterozygous 1 bp, 10 bp, and 654 bp *COL7A1*-deletions were detected via WGS coverage (right; see Supplementary Data 3) in iPS cells/iSCs from homozygous patients. **h** Variants identified via the STAMPv2 oncopanel. Germline variants (green) are found in all 3 cell types. A heterozygous androgen receptor mutation (red) stems from clonal expansion of a fibroblast subpopulation (3%) with this lesion. Blue: variants identified by secondary methods. Source data are provided as a Source Data file.

C8, which were present in the other three lines (Figs. 3i, j and 5c). Moreover, graftability correlated with the Gibbin-dependent dermal gene signature[28] (Fig. 5d, Supplementary Fig. 7, and Supplementary Data 1). To independently verify this finding, we measured the amount of Gibbin-dependent mesoderm in four iSC lines (*n* = 3–7) by flow cytometry using the Gibbin-dependent marker CD90/Thy1 (Supplementary Fig. 7a, e and Fig. 5e, f). Again, the two low-efficiency lines exhibited few CD90+ cells whereas the better-performing lines consistently produced 2–5% CD90+ dermal cells (Fig. 5e, f). Thus, optimal graftability correlated with the presence of Gibbin-dependent dermal cells, which have been shown to provide a critical maturation signal for epidermal stem cells[28]. In addition, these results nominate another product potency marker, i.e. CD90/Thy1 (along with ITGA6). Further studies are necessary to determine a deeper mechanistic insight into how clusters C5/C8, and potentially Gibbin-dependent fibroblasts, program successful stratification and grafting of iSCs. Concerning the latter, we assayed dermal cells of murine graft areas for binding of human-specific antibodies, including vimentin and human nuclei. These data found that cells from the human iSC graft reside in the dermal layer and are positive for the fibroblast-marker vimentin. Our analyses indicate that the different cell types of the iSC assume their correct physiological localization, potentially improving the stratification of the graft (Supplementary Fig. 9a–d).

The main safety concern of the iSC product is tumor formation, as squamous cell carcinomas (SCC) are a common complication within the chronic wounds of adult EB patients. Careful investigation failed to show any histopathological signs of SCC formation in any of the human organoid skin grafts (Fig. 5h). To assess the sensitivity to detect potential tumors in this assay, we performed spike-in positive control experiments (Supplementary Fig. 10a, b). To address the possibility of potentially metastasizing tumor cells from graft sites we designed a method to detect human DNA by Alu sequence qPCR from 7 organs and blood from all grafted mice at 1, 3, 6, and up to 9 months (Fig. 5g–i)[48,49]. Spike-in experiments using 1–10,000 iSCs in 500 K mouse cells from various organs demonstrated the level of detection (LOD) as 3000 iSCs for lymph node, brain, and liver. Sensitivity for quantitation was determined to be 10,000 iSCs at 38 PCR cycles for all four organs tested (Supplementary Fig. 10c). Alu-qPCR results for DEB125-1 iSC grafted mice at 1 and 6 months showed no detection of human Alu sequences in all tested mouse organs or blood, with all iSC grafts tested. Subsequently, the four patient lines, grafted on mice, were all assayed for Alu-qPCR with no detection (Fig. 5i and Supplementary Fig. 10d).

Another potential DEBCT-associated tumor risk is teratoma formation from residual undifferentiated iPS cells. Thus, we evaluated pluripotency marker expression in expanded ITGA6-enriched iSCs. *LIN28A*, *NANOG*, *OCT4*, and *SOX2* were below detectability in iSCs of five different cell lines (Fig. 5j). Utilizing RNA expression of the

pluripotency marker *LIN28A*, we measured the LOD of contaminating iPS cells at 0.38% (i.e., 1 iPS cell in 260 immortalized keratinocytes[50]; Supplementary Fig. 10e). In combination with absence of ITGB4 in iPS cells, which is necessary for adhesion to the BMZ[32,33], we predicted the likelihood of teratoma formation to be low. Indeed, no teratomas were found in grafted animals. However, to analyze potential biodistribution of a RDEB patient iPS cell-derived teratoma, we subcutaneously injected immunocompromised mice with two RDEB iPS cell lines (i.e., CO1-131 and DEB125-1). After the expected formation of teratomas at the injection sites, we failed to detect human DNA in all other organs tested (Supplementary Fig. 10f). This indicates that residual undifferentiated iPS cells contained in the DEBCT product are unlikely to disseminate. In sum, our data reinforces the safety and efficacy of our cell manufacturing method to the level of detectability.

## Discussion

With DEBCT manufacturing, we overcome existing technical hurdles to develop a scalable, cGMP-compatible, and efficient platform for the derivation of autologous and genetically corrected organotypic skin grafts for definitive closure of RDEB patient wounds. This platform combines the generation and genetic correction of patient-derived iPS cells in a single manufacturing step and details a strategy for safe and reproducible production of graftable organotypic skin composites at clinical scale. The therapeutic product is composed of basal keratinocytes, dermal fibroblasts, and melanocytes, better resembling the composition of physiological tissues than previous approaches. Moreover, our use of generalizable manufacturing reagents and development of efficacy, toxicology and product characterization assays provide reproducible regulatory and manufacturing paths.

An important aspect of this study is the successful combination of CRISPR/CAS9-mediated gene editing and reprogramming into a single manufacturing step. Previous approaches generated iPS cells to be genetically corrected in a subsequent clonal step, necessitating longer manufacturing time, higher cell expansion rates, and extensive additional quality control and release tests. A methodological combination of reprogramming and gene editing was possible due to a series of advances. First, several reagents enable clinical-scale corrected iPS cell manufacturing. An advanced transfection reagent allows efficient delivery of three different nucleic acid/protein cargos with low toxicity, the employed mRNA-mediated reprogramming kit derives iPS cells with high efficiency, and a chemically defined iPS cell expansion media ensures genomic/chromosomal stability. Second, the *COL7A1* correction and iPS cell reprogramming steps are seamless and lack genomic alterations other than the corrected mutation and designed silent point mutations that facilitate genotyping. Hence, our approach removes potential complications associated with most other gene and cell therapy approaches, i.e. random insertion mutagenesis, introduction of non-physiological gene regulatory elements, or alteration

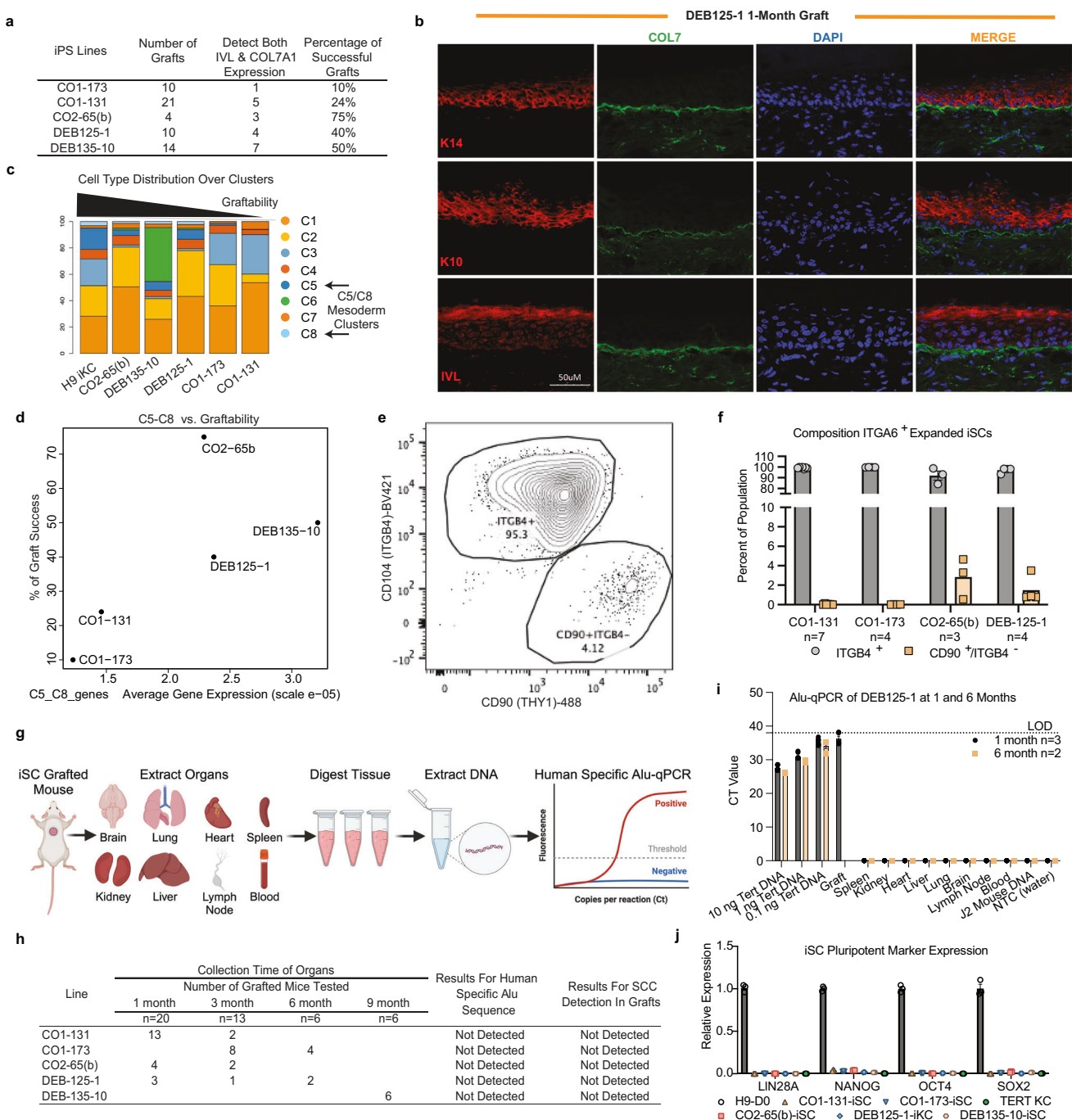

**Fig. 5 | Patient iPS cell-derived organotypic skin grafts survive in mice with a favorable safety profile. a** Summary of mouse graft success determined by IF staining of basement marker Collagen 7 and granular marker involucrin (IVL) at 1 month. **b** Representative IF image of DEB125-1 derived iSC graft at 1 month. Keratin K14, K10, Involucrin (red), Collagen 7 (COL7; green), DAPI (blue). **c** Quantification (%) of 8 different cell-type clusters (C1–C8) comprising the DEBCT product and identified via individual scRNA sequencing (see Fig. 3). **d** Strong positive correlation between % graft success rate (**a**) and average C5/C8 cluster expression as quantified by scRNA sequencing. **e** Representative flow cytometry plots analyzing cell composition of DEB125-1 iSCs labeled for ITGB4 and CD90/Thy1 before grafting onto mice. **f** Quantification via flow cytometry of the average expanded iSC cell composition prior to grafting (*n* = number of biological replicates as indicated; mean and SEM are shown). **g** Overview of method for detection of evading iSCs into mouse organs using human-specific Alu-qPCR. **h** Table summarizing Alu-qPCR-based biodistribution and histology-based tumor detection results from mice at 1, 3, 6, and 9 month post iSC grafting. **i** Representative Alu-qPCR from organs of DEB125-1 derived iSC grafted mice at indicated time points. LOD is level of detectability in tissues spiked with human DNA from TERT keratinocytes (*n* = number of biological replicates as indicated; mean and SEM are shown). **j** qRT-PCR detection of pluripotency marker expression (*LIN28A, NANOG, OCT4* and *SOX2*) in the iSC product. H9 ES cells and TERT keratinocytes (KC) were used as controls (*n* = 3 technical replicates; mean and SEM are shown). Source data are provided as a Source Data file. Panel **g** created with BioRender.com released under a Creative Commons Attribution-NonCommercial-NoDerivs 4.0 International license.

of endogenous loci. Third, under optimized conditions genetically corrected autologous iPS cell banks can now be obtained and characterized in less than a month after dermal punch biopsy. This substantial acceleration not only increases the cells safety profile by lowering culture-induced mutation burden[8], but also reduces the time/cost of autologous iPS cell manufacturing. In comparison to off-the-shelf allogeneic iPS/ES cell-derived methods, our autologous approach also does not suffer from host tolerance issues[51].

We discovered one important caveat of CAS9-mediated editing. Whenever we observed integration of ssODN sequences on one allele, we almost always found InDel mutations on the other, which agrees with error-prone non-homologous end-joining (NHEJ) being the predominant repair mechanism for DNA breaks in human cells, The exception are the heterozygous compound mutation carriers, in which the other allele is not an on-target[52]. These InDels can be very large (Fig. 2), resulting in standard PCR genotyping failing to amplify the affected allele and giving the false impression that genetic correction was bi-allelic. Our quantitative ddPCR analysis indicated that of 479 iPS cell lines derived from homozygous patients, only 3 exhibited potential bi-allelic integration of donor sequences (i.e. <1%; Supplementary Fig. 4). This contrasts with reports of significantly higher bi-allelic *COL7A1* editing events in primary RDEB keratinocytes and iPS cells[22,53]. While we cannot exclude that primary keratinocytes and iPS cells may exhibit more favorable NHEJ to ssODN-integration ratios, we note that reported claims of up to 100% bi-allelic correction are based on genotyping by non-quantitative PCR, which could have produced false negative results in case of CAS9-induced large deletions at the non-corrected allele[54–60].

Our cell manufacturing method for deriving engraftable organotypic iSCs is tailored for scalable cGMP-compatible production in a 45-day process under optimal conditions. Used materials and methods are defined, xeno-free, extensively validated[30,61], and overcome three key cell manufacturing hurdles. First, the induction of surface ectoderm, mesoderm and neuroectoderm mimics developmental signaling required for proper tissue maturation including development of epidermal, dermal, and melanocyte precursors. We find a significant proportion of the final iSC product to be the holoclone-like population previously identified to be responsible for long-term keratinocyte maintenance[39,40]. In contrast to the previous culture-intensive methods of subcloning and the use of a non-reproducible mouse fibroblast feeder line for maintenance, our inductive method produces all cell types necessary for an organotypic therapeutic product in scalable quantities via a defined and xeno-free method. In addition, our studies support our previous results that underline the importance of epidermal-mesodermal signaling for subsequent induction of the region-dependent epidermal stratification program[28], as cell lines deficient in differentiating into the Gibbin-dependent dermal population had a lower grafting efficacy. The reason for this line-to-line variability remains undetermined but the identification of the Gibbin-dependent CD90 surface marker, expressed on a small subpopulation of mesodermal-like iSC cells, provides a quantifiable biomarker that may predict efficacy of graftability. Future advancements may include the identification of distinct mesodermal signals that could give rise to distinct epidermal stratification programs, facilitating more subtle tissue morphologies with minimal morphogen reagent concentrations.

Second, incorporation of the ITGA6-enrichment step overcomes maturation heterogeneity that reduces epidermal polarity and graft stratification. Despite a defined multi-lineage organoid culture aimed to regulate epidermal−dermal induction, immature cellular products appear to act dominantly to reduce polarity and stratification abilities, thus requiring their removal for optimal grafting performance. While the differentiation efficiency varied between patient clones (and even iPS cell lines without known genetic pathogenicities, Supplementary Fig. 6d), our development of an ITGA6-enrichment step allowed

derivation of organotypic skin grafts at clinical scale from all patients included in the study. Cell sorting technology for enriching antigen-specific cells is being used in various approved clinical products, including CAR T cell and CD34+ cell enrichment[62]. Because ITGA6 and its binding partner ITGB4 are present at elevated levels on most epithelial stem cells, our enrichment method will be applicable to other epithelial tissues.

Third, iPS cells are not subjected to the Hayflick limit and provide a source for virtually unlimited production of iSC grafts, circumventing problems arising from paucity of expandable keratinocytes or holoclones exhibited by some patients. In a recent study, $1.5 \times 10^9$ patient-derived cells transduced with Moloney virus conferring expression of LAMB3 cDNA were used to graft $0.85\,m^2$ of body surface from a junctional EB patient[14]. Given an average coupling efficiency of ~75%, we would need to culture $2 \times 10^9$ iPS cells to manufacture $1.5 \times 10^9$ iSCs, which is in the realm of current technical feasibility.

While our abbreviated manufacturing process greatly reduced the number of cell doublings and consequently the chance of culture-induced mutations[63], our detailed characterization by whole and targeted genome sequencing, paired with animal toxicology, provides a rigorous assessment of manufacturing-associated mutational and pathological risks. A key question was whether genetic variants in the therapeutic product are introduced during cell manufacturing or were pre-existing in patient skin cells. Sensitive ddPCR and WGS techniques confirmed that many variants stem from clonal amplification of pre-existing somatic mutations rather than de novo manufacturing-induced mutations. In agreement with high genomic stability within our manufacturing process, we found no variants unique to iPS cells only. We note that in line with naturally occurring proliferation-induced random mutagenesis[63], a small fraction of all detected variants arises de novo, as indicated by rare iSC-specific variants not found in parental fibroblasts and clonal parental iPS cell lines (Fig. 4c and Supplementary Fig. 8). We did not observe any overlap of such iSC-specific variants between lines derived from different individuals or the same patient. This reflects the random nature of naturally occurring somatic mutations and legitimates our advancement that now allows DEBCT manufacturing in an unprecedented short time, thereby minimizing the risk of introducing a haphazard mutation with deleterious consequences. We did observe a threefold line-to-line variability in the total number of SNVs arising in the different iPS cell lines (~1300 to ~3900). However, since many variants arise from clonal amplification of pre-existing somatic mutations, we conclude that this variation derives from the different variant load of patient fibroblast.

Only three variants in only one detection method (k-means clustering) were found to be shared between patients and all of them were pre-existing. Similar results were found by the CLIA-certified STAMPv2 next-generation sequencing assay to detect cancer-driving variants. In line with our sequencing analysis, sensitive toxicology assays for residual pluripotent or carcinogenic cellular components found no evidence for local or distant invasive properties of the product. This included grafts derived from patient cells harboring a mutation in the androgen receptor, which scored positive on the STAMPv2 screen. We conclude that the STAMPv2 variant detection assay, along with our toxicology assays, will ensure a well-defined cellular product with low risk.

## Methods
### Human subjects
This study was compliant with all regulations and was approved by Stanford University SCRO protocol #691 and human subjects IRBs # IRB-45005, IRB-22237, and IRB-22005.

### Derivation and culture of primary patient fibroblasts
Fibroblasts from patients DEB125 and DEB135 were derived from a fresh dermal punch biopsy. The anatomical localizations of dermal

punch biopsies are as follows: patient DEB125−left thigh; patient DEB135−right hip. Minced pieces (~1 mm³) of biopsies were cultured in DMEM (Gibco 12-430-062) with 10% fetal bovine serum (FBS; Hyclone SH30406.02 New Zealand sourced) and 1% Penicillin-Streptomycin (Fisher Scientific 15140163). Media was changed every 4 days and fibroblasts started to grow out from the tissue at day 4 (DEB125) and 8 (DEB135), respectively. Fibroblasts were trypsinized (TrpLE; Gibco A1285901) and stored in Cryostore CS10 (StemCell Technologies 7930) in the vapor phase of liquid nitrogen. Fibroblasts from patients CO1 and CO2 were a generous gift from Dr. Dennis Roop (University of Colorado) and cultured in the above media. Patient material used in this study has been provided de-identified. Karyotyping revealed that of four patients whose cells were used in this study, two are genetically male, and two are genetically female, with no other karyotypic abnormalities. This reflects a balanced distribution of gender for this study.

## Single-step editing/reprogramming

Immediately after transfection of patient fibroblasts with RNPs and ssODNs (see below), cultures were expanded to 225k cells in order to reprogram 3 wells of a six-well dish. iPS cells were induced with a reprogramming kit, a generous gift of iPEACE Inc (Los Altos, CA) according to the manufacturer's recommendations. Briefly, 75k cells were seeded per well of a six-well plate previously coated with iMatrix (Reprocell NP892-012) and transfected (VFS2205; see below) with 150 ng of reprogramming mRNAs (iPEACE Inc) for 10 consecutive days. Cultures were subsequently maintained in StemFit media (Nacalai USA Basic03) according to the manufacturer's recommendations (StemFit was supplemented with basic FGF from Preprotech AF-100-18B and Rock inhibitor from Axon 1683). After the emergence of iPS cell colonies, non-reprogrammed cells encasing iPS cell colonies were removed as outlined in Supplementary Fig. 3a−c and remaining iPS cells were incubated for an additional 24 h before manual picking of colonies. After picking of 1-step edited/reprogrammed iPS cells (i.e. passage (P)0) and subsequent culture in iMatrix-coated 48-well plates, we duplicated the lines into sister plates (P1). P1 iPS cells were screened for *COL7A1* editing via ddPCR and positive hits from the sister well were passaged one more time (P2) for expansion and biobanking. P2 iPS cells were then thawed, expanded, and subjected to iSCs-differentiation at P4. For karyotyping and NGS analysis, P2 iPS cells were thawed and expanded to at least $30 \times 10^6$ cells. Our 1-step editing/reprogramming methodology does not use any pharmacological selection. iPS cells were qualified as described in the manuscript (expression of multiple pluripotency markers, karyotyping, etc.). Our empirical observation is that our culture system, i.e. StemFit media from Ajinomoto, does not support continuous passaging and culturing of cells other than iPS cells (e.g. primary fibroblasts).

## iPS and H9 ES cell differentiation

Established iPS cell lines with Certificate of Analyses were WTC-11 (https://hpscreg.eu/cell-line/UCSFi001-A) and AICS0017 (iPS cell-DSP, Coriell). Established and study-generated iPS cell lines were maintained in StemFit Basic03 media (Ajinomoto). After expansion, the cells were prepared for embryoid body formation using the Aggre-Well™ EB 400 microwell plate following the manufacturer's recommendations (StemCell Technologies). Cells were dissociated in Accutase (Innovative Cell Technologies) counted and added at 1.2 million cells per microwell in AggreWell media containing 1 μM ROCK inhibitor (StemCell technologies 72302). The media was carefully changed the following day to remove ROCK inhibition. Embryoid bodies were collected after 48 h and plated on 10-cm plates at roughly 250 Embryoid bodies per plate in StemFit Basic03 media until cell attachment on vitronectin (Gibco™) pre-coated plates following the manufacturer's recommendations. To initiate differentiation of pluripotent stem cells (PSC) to keratinocytes, the PSCs were induced with

Essential 6 media (Gibco™) containing 5 ng/mL BMP-4 and 1 μM retinoic acid for 7 days, followed by culture in Defined Keratinocyte Serum Free Medium (DKSFM, Gibco™) through day 45. The same media was continued after MACS enrichment. The enriched keratinocytes were seeded onto Corning PureCoat™ ECM Mimetic 6-well Collagen I Peptide Plates (Corning). H9 ES cells (WiCell) were maintained on matrigel hESC qualified matrix (Corning) and in Essential 8 media (Gibco™) until differentiation began as described above.

## ITGA6 enrichment by Miltenyi AutoMACS and CliniMACS plus separation

Differentiated day 45 cells were dissociated with Accutase (Innovative Cell Technologies) for 30 min, washed, and counted in 10−20 mL of wash buffer containing PBS (Gibco), 1 μM EDTA (Lonza), 2% BSA (Miltenyi) and 1 μM ROCK inhibitor (StemCell technologies 72302). For all wash steps, cells were pelleted in the wash buffer at 1000 rpm for 5 min. Cell pellets were resuspended in FcR Blocking reagent (Miltenyi) at 20 μL FcR to 80 μL of wash buffer for up to $1 \times 10^7$ cells for 5 min at room temperature. CD49f biotin antibody (REA518, Miltenyi), was added to the blocked cells at 1:50 dilution and incubated for 20 min at room temperature. After incubation, the wash buffer was adjusted to 10 mL and cells were pelleted at 1000 rpm for 5 min. After aspiration of wash buffer, cells were resuspended in 80 μL of buffer and 20 μL of Anti-biotin IgG microbeads (Miltenyi Biotec) and incubated at room temperature for 15 min. Cells were then washed as previously described and resuspended in fresh wash buffer up to 4 mL. The labeled cell suspension was MACS separated using the AutoMACS (Miltenyi Biotec) PosselD program setting. For CliniMACS Plus cell enrichment, the protocol was modified slightly. Cells were labeled as described using the REA518 antibody. The Anti-biotin reagent (microbead) was used according to the manufacturer's recommendation at 1 mL of microbead to 12.5 mL of CliniMACS Plus buffer with 2% HSA and 1 μM ROCK inhibitor. Three CliniMACS Plus program settings were tested to optimize cell purity, recovery and viability using the CliniMACS LS Tubing set: program 1.1 (gentle), program 5.1 (higher purity), and program CD34.1 (lower purity, higher viability). The optimal program found was CD34.1. Following separation, the cells were resuspended in Defined Keratinocyte Media (Gibco) and plated on ECM Collagen I coated peptide plates (Corning).

## Production of RNPs

RNPs were generated by mixing sgRNAs with recombinant CAS9 at a molar ratio of 6:1, followed by incubation at room temperature for 30 min. Briefly, to generate 100 μL with a concentration of 0.5 pmol RNP/μL we combined 81.13 μL PBS with 18 μL sgRNA (650 ng/μL H₂O) and 0.87 μl CAS9. CAS9s were sourced from Integrated DNA Technologies (IDT; Alt-R® S.p. CAS9 Nuclease 1081058 or Alt-R® S.p. HiFi CAS9 Nuclease 1081060) and Aldevron (SpyFi CAS9 Nuclease 9214-0.25MG). sgRNAs were sourced from Synthego and had the following sequences fused to an 80-mer SpCAS9 scaffold: C1 GGAUCCACCGUG AGUCCUCG, C2 GGGAUCCACCGUGAGUCCUC, C3 CGGGAUCCACCG UGAGUCCU, C4 ACUCACGGUGGAUCCCGCUG, C5 GACUCACGGUGG AUCCCGCU, C6 GGACUCACGGUGGAUCCCGC, and DEB135 ACUGG CACCAUCUCAACCUG.

## ssODNs

ssODNs were sourced from IDT and stored as 10 μM stock aliquots in H₂O at −20 °C. The ssODNs used to edit the Colorado mt allele covered the genomic sequence of chromosome 3 between coordinates 48,570,186−48,569,987 (GRCh38; https://asia.ensembl.org/) and included the 4 silent mutations described in Fig. 1b. The ssODNs used to edit fibroblasts of patient DEB135 covered the genomic sequence of chromosome 3 (GRCh38; https://asia.ensembl.org/) between the following coordinates and included the four silent mutations described in Supplementary Fig. 5a: ssODN 84 bases 48,572,943−48,572,860; ssODN

127 bases 48,573,004–48,572,878; ssODN 200 bases 48,572,943–48,572,744.

## Transfection with RNPs and ssODNs

One day before transfection, 25k fibroblasts were seeded in penicillin-streptomycin-free media (see above) in a well of a 24-well plate. Transfection of 5pmol RNP (or of amounts indicated in Supplementary Fig. 1d) and 10pmol ssODN was performed with VFS2205 transfection reagent (Vivofectamine™ Services from Thermo Fisher Scientific) according to the manufacturer's recommendations.

## Extraction of genomic DNA

Genomic (g)DNA was extracted using the Quick DNA miniprep kit (Zymo Research D3025) according to the manufacturer's recommendations. Genomic DNA was eluted and stored in nuclease-free H2O (Ambion AM9937).

## TIDE assay

TIDE assay[64] was performed as per the inventor's instructions via the online tool at https://tide.nki.nl/. TIDE plots were prepared using Adobe Photoshop CS6 (64 bit). Briefly, edited loci were amplified from gDNA extracted 3 days (or 7 days for Fig. 4e, g) after transfection with RNPs and ssODNs, using the primers outlined in the Supplementary Information and Q5 Hot-start high-fidelity polymerase (New England Biolabs M0494S). PCR products were extracted from standard agarose gels using the QIAquick Gel Extraction Kit (Qiagen 28704) and sequenced by Elim Biopharm (www.elimbio.com).

## ddPCR

gDNA was used as a template for droplet digital (dd) PCR according to the manufacturer's instructions for the BioRad QX200 system. ddPCR reactions were prepared using 2× ddPCR supermix (BioRad 1863025), bi-allelic reference-HEX primer/probe mix (primer 1: GGATGGG-GAATGCAGCTCTT, primer 2: AGTGCGGCAGAATACAGCA, probe:5′-HEX/TGATGGGTT/ZEN/GTGAAGGCAGCTGCACCT/3′IABKFQ), and one of the following FAM-conjugated primer/probe mixes: edited Colorado mt allele-FAM (primer 1: GAGTCAATGAACCTAATGTC, Primer 2: AGAGAGTCCTGGGGTA, probe: 5′6-FAM/AAGGGGGCT/ZEN/CACGGTG/3′IABKFQ), or edited DEB135 mt allele-FAM (primer 1: GACAGAGCTCTTCCCTCTCA, primer 2: CTGCCCCCAGAACACATAC, probe: 5′6-FAM/TGGCCGAGA/ZEN/CGGTGCC/3′IABKFQ). After generation of droplets in the BioRad QX200 generator, which employed DG8 cartridges (BioRad 1864008), gaskets (BioRad 1863009), and droplet generation oil for probes (BioRad 1863005), ddPCR mixes were loaded into 96 well plates (Fisher Scientific E951020346) and sealed with pierceable foil heat seal (BioRad 1814040). PCR reactions were run in a thermocycler using the following parameters: $1 \times 10$ min at 95 °C, $47 \times 30$ s at 94 °C followed by 1 min 5 s at 57 °C, and $1 \times 10$ min at 98 °C. Subsequently, ddPCR reactions were analyzed in a BioRad QX200 droplet reader and data analysis was performed with BioRad's QuantaSoft Analysis Pro 1.0.596 software or QuantaSoft software (v1.7.4.0917) and Microsoft Office Excel. ddPCR plots were prepared for publication using Adobe Photoshop CS6 (64 bit). The percentage of edited *COL7A1* alleles was calculated via the following Eq. (1): ((concentration in copies per μL of edited Colorado or DEB135 allele [FAM signal]*100)/concentration in copies per μL of bi-allelic reference [HEX signal])*2 = % heterozygously edited cells. For the competitive ddPCR assay detecting the wt and mt (c.1427 G > A, p.G476E) allele of the androgen receptor (*AR*), following primer/probe mixes were employed: primer 1: GAAGGCCAGTTGTATGGAC, primer 2: CACAT-CAGGTGCGGTGAAG, *AR*-wt-FAM: 5′6-FAM/AGGCGGGAG/ZEN/CTGT AGCCC/3′IABKFQ, *AR*-mt-HEX: 5′HEX/AGGCGGAAG/ZEN/CTGTAGCCC C/3′IABKFQ. Competitive ddPCR assays were run as above with the following thermocycler parameters: $1 \times 10$ min at 95 °C, $45 \times 30$ s at 94 °C followed by 1 min 5 s at 56 °C, and $1 \times 10$ min at 98 °C. The percentage of *AR*-mt alleles was calculated via the following Eq. (2): ((concentration in copies per μl of *AR*-mt [HEX signal]*100)/concentration in copies per μL of *AR*-mt [HEX] + *AR*-wt [FAM])*2 = % cells with *AR*-mt. All ddPCR primer/probe mixes were sourced from IDT as PrimeTime Std qPCR Assays with a primer/probe ratio of 3.6.

## Topo cloning

Topo cloning of PCR-amplified *COL7A1* alleles was achieved using the Zero Blunt TOPO PCR Cloning Kit with One Shot TOP10 Chemically Competent *E. coli* cells (Thermo Fisher Scientific K2800-40) according to the manufacturer's recommendations.

## Sanger sequencing

Sanger sequencing was performed at Elim Biopharm according to the provider's recommendations. Sequencing primers for Topo-cloned PCR products had the following sequences: Fw 5′-GTAAAAC-GACGGCCAG-3′, Rw 5′-CAGGAAACAGCTATGAC-3′. After purification from standard agarose gels (QIAquick Gel Extraction Kit, Qiagen 28704), PCR products were sequenced with either the Fw and/or Rw primer listed in the Supplementary Information. Sanger sequencing data provided by Elim Biopharm was analyzed using Applied Biosystems Sequence Scanner Software 2, Microsoft Office Word, sequence alignment software tools available at https://blast.ncbi.nlm.nih.gov/Blast.cgi, and Adobe Photoshop CS6 (64 bit).

## In silico analysis of sgRNAs

sgRNA sequences were analyzed via the CRISPOR algorithm in order to identify activity and specificity scores and off-targets (http://crispor.tefor.net/crispor.py)[65,66].

## PCR

PCRs were performed with the Q5 Hot Start High-Fidelity 2× Master Mix (New England Biolabs M0494S) and primers listed in the Supplementary Information. *E. coli* colony PCRs (Fig. 1f and Supplementary Fig. 1f–h) were performed with CloneID 1× Colony PCR Mix (Lucigen 30059-2) and primers listed in the Supplementary Information. Agarose gels for analysis of PCR products included the 1 Kb Plus DNA Ladder (Thermo Fisher Scientific 10787018) as a size reference. Data from agarose gels were collected using a BioRad Molecular Imager Gel Doc XR+ imaging system with associated software, i.e. Image Lab 5.1. Images of gels were processed using Adobe Photoshop CS6 (64 bit). Uncropped versions of images are provided in source data.

## Immunofluorescence microscopy

Cells were fixed (4% paraformaldehyde), permeabilized (0.2% Triton X-100; except for Fig. 2h−TRA-1-81), blocked (BSA), and stained with antibodies as per standard laboratory procedures. Nuclei were counterstained with DAPI. Primary antibodies used: TRA-1-81 (Sigma Aldrich MAB4381, 1:1000), TRA-1-60 (Sigma Aldrich MAB4360, 1:1000), NANOG (Abcam ab21624, 1:2000), K18 (R&D AF7619, 1:800), K14 (BioLegend SIG-3476-100, 1:800), p63 (Gene Tex GTX102425, 1:100), ITGA6 (Millipore MAB1378, 1:200), Involucrin (Abcam ab27495, 1:100), K14 (Covance PRB-155P, 1:2000), K18 (Cell Signaling 4548, 1:400), K10 (Covance PRB-159P, 1:500), human-specific C7 LH7.2 (Millipore MAB1345, 1:250), Human Nuclear Antigen Antibody (Thermo Fisher RBM5-346-P1, 1:200), Vimentin [RV203] (Abcam ab8979, 1:100), and CD104/Integrin beta 4 (Thermo Fisher 14-1049-82, 1:50). Immunofluorescence microscopy data was collected using a Leica DMi8 with associated software, i.e. Leica Application Suite X 3.7.4.23463, a Carl Zeiss Axio Observer.Z1 inverted microscope with Axiovision software, or a Leica TCS SP5 confocal laser scanning microscope with associated software, i.e. Leica Application Suite for Advanced Fluorescence software v.2.7.9. Tissue sections were co-stained with Hoechst and slides were mounted with the Prolong Gold mounting medium (Life

Technologies). Images were processed using Adobe Photoshop CS6 (64 bit) or Adobe Illustrator v26.

## Karyotyping
Karyotyping was performed at WiCell or the Stanford University Medical Center Cytogenetics laboratory after expansion of iPS cells to at least 30 million cells.

## Genomic DNA extraction and whole-genome sequencing
Genomic DNA (gDNA) was extracted from cell pellets of skin fibroblasts, iPS cells, and derived iSCs from three patients (11 total samples) with the MasterPure Complete DNA Purification Kit (Lucigen #MC85200). The gDNA yield was quantified by the Qubit dsDNA high-sensitivity (HS) fluorescence assay (Invitrogen # Q32851) and ranged from 39.6 to 106.0 ng/μL in a final volume of 25 μL. The gDNA purity was verified by the NanoDrop 2000 spectrophotometer (Thermo Scientific #ND-2000). Tagmentation-based sequencing libraries were prepared from 500 ng gDNA in duplicate with the Illumina DNA prep (M) kit (Illumina #20018704) using IDT® for Illumina® DNA/RNA UD Indexes Set B, Tagmentation (Illumina #20027214). Libraries were sequenced in 150-bp paired-end format on two sequential NovaSeq 6000 S4 lanes (Novogene Corporation Inc.). Each sample obtained at least 40X average coverage after combining the data from both library replicates. See Reporting Summary for details.

## Read pre-processing and variant-calling
Raw reads were trimmed of adapter sequences using cutadapt[67] in pair-end mode and subsequently mapped to the human hg38 reference genome (https://hgdownload.soe.ucsc.edu/goldenPath/hg38/bigZips/hg38.fa.gz) using BWA-MEM[68]. The BAM files were processed by GATK4[69] to sort mapped reads and mark PCR duplicates, followed by base quality score recalibration (BQSR) which utilized files of known human variant sites recommended in the best practice workflow of GATK4. SNPs and/or Indels were called from processed BAM files using four variant-calling algorithms separately: HaplotypeCaller[69,70], Mutect2[71], Lofreq[72] (SNP only), and Scalpel[73] (Indel only). The overlapping hits of three SNP callers (HaplotypeCaller, Mutect2 and Lofreq2) and of three Indel callers (HaplotypeCaller, Mutect2 and Scalpel) were regarded as true variants. Functional annotations of the .vcf files were generated using ANNOVAR with the embedded refGene protocol for hg38. Only variants that do not match the dbSNP database[74] (build 138) were considered and were included in subsequent analyses. See Reporting Summary for details.

## K-means classification and cell-type-specific variant analysis
For each patient-derived lineage, mutations called in skin fibroblasts, iPS cells or derived iSCs were merged into one table with allele frequency (AF) information retained in all three cell types. Each variant $X$ then has its AF information encoded as vector of three elements via the following Eq. (3):

$$\overline{AF}_X = (AF_{X,\text{fibro}}, AF_{X,\text{iPS cell}}, AF_{X,\text{iSC}})$$

Unsupervised K-means clustering was conducted with all considered mutations using function "kmeans" in R package "stats". Pseudo seeds were set for the random initiations of clustering centroids to ensure reproducible classification outcomes. Max iteration number was set as 1000, and "Lloyd" was selected as the clustering algorithm. The pre-defined cluster number $K$ was tested from $K = 3$ to $K = 9$ in each individual task. The results of different patients consistently reveal a cluster exhibiting iPS cell/iSC-specific pattern at $K = 9$ as highlighted in Supplementary Fig. 8b. Therefore, we picked the smallest $K$ numbers in each individual that defines such pattern (i.e., $K = 8$ for CO1-131 and CO1-173; $K = 4$ for CO2-65(B); $K = 9$ for DEB125-1), and generated variant lists from the relevant clusters to be used in

Supplementary Fig. 8c. To further identify variants that are specifically represented in iPS cells and/or iSCs compared with their parental fibroblasts, we defined odds ratios OR to measure over-representation of a mutated allele in one cell type relative to another. For example, the odds ratio for variant $X$ to be specifically represented in iPS cells but not the fibroblasts is calculated via Eq. (4):

$$OR_{X,\text{iPS cell/fibro}} = \frac{AF_{X,\text{iPS cell}} \times (1 - AF_{X,\text{fibro}})}{AF_{X,\text{fibro}} \times (1 - AF_{X,\text{iPS cell}})}$$

Any variants $X$ passing the below filters were selected as iPS cell and/or iSC-specific variants shown in Fig. 4c:

1. Shared iPS cell/iSC-specific:

$$\left(AF_{X,\text{fibro}} < 0.25\right) AND \left(AF_{X,\text{iPS cell}} > 0.25\right) AND \left(AF_{X,\text{iSC}} > 0.25\right)$$
$$AND \left(OR_{X,\text{iPS cell/fibro}} > 2.0\right) AND \left(OR_{X,\text{iSC/fibro}} > 2.0\right);$$

2. iPS cell-specific:

$$\left(AF_{X,\text{fibro}} < 0.25\right) AND \left(AF_{X,\text{iPS cell}} > 0.25\right) AND \left(AF_{X,\text{iSC}} < 0.25\right)$$
$$AND \left(OR_{X,\text{iPS cell/fibro}} > 2.0\right) AND \left(OR_{X,\text{iPS cell/iSC}} > 2.0\right);$$

3. iSC-specific:

$$\left(AF_{X,\text{fibro}} < 0.25\right) AND \left(AF_{X,\text{iPS cell}} < 0.25\right) AND \left(AF_{X,\text{iSC}} > 0.25\right)$$
$$AND \left(OR_{X,\text{iSC/fibro}} > 2.0\right) AND \left(OR_{X,\text{iSC/iPS cell}} > 2.0\right)$$

## Coverage analysis of potential off-target sites and homology search with used sgRNA C4
The 1KB and 1MB up- and downstream regions of the sgRNA C4-targeted site (COL7A1), as well as 57 predicted off-target sites, were investigated for potential alterations led by sgRNA-dependent effects (Fig. 4f). Normalized coverages were calculated as absolute read depth per site divided by the mean coverage of the entire chromosome. Homology search between variant sites and the sequence of sgRNA C4 was conducted using a custom script available at our GitHub repository: https://github.com/shli-embl/hg_wgs_variant_calling/.

## GO-enrichment analysis
GO-enrichment analysis was performed via the online tool at http://geneontology.org/[75,76].

## Next-generation amplicon sequencing
Samples were harvested at 3 and 14 days after transfection with RNPs and ssODNs. Genomic DNA was purified using the Quick DNA miniprep kit (Zymo Research D3025) according to the manufacturer's recommendations and used as a template for PCR using the following primers: Fw 5′-CCTCTGAGTCAATGAACCTAATG-3′ and Rw 5′-CTACAGGAACCAGGGCAGTG-3′. Four technical replicates were produced per group (i.e., sgRNA C2, sgRNA C4, and control) at each time point and amplicons were sequenced by Novogene USA on the NovaSeq X Plus lane with the 10B flow cell. Subsequently to demultiplexing the raw reads, we used CRISPResso2[77] to analyze and quantify integration of donor sequence and InDel events within each sample dataset. The program was executed in batch mode, with the expected edited sequence indicated by parameter "-expected_hdr_amplicon_seq". Read counts for each subclass of editing outcomes were extracted from final reports. To account for imperfect donor integration events, where some but not all template-encoded variants are installed, we extended the quantification window which is by default a 2-bp window centered at the cleavage site. This extension was achieved by redefining two parameters "-quantification_window_center" and "-quantification_window_size", to ensure the

quantification window overlaps all intended variants on the template. This customized step allows the program to precisely quantify the percentage of amplicons that perfectly match the designed template, categorized as "complete donor integration", whereas other events with mismatches are classified as "incomplete donor integration" or "ambiguous". (The latter are sequences that align equally well to both, the expected repaired allele or wild-type sequence).

Primers to generate the PCR amplicons for the biological replicates shown in Supplementary Fig. 2g, h are as follows: Fw 5′-CCTCTGAGTCAATGAACCTAATGTC-3′ and Rw 5′ -TGAGCTACAGGAACCAGGGCAG-3′. These samples were sequenced using the Amplicon-EZ service from Genewiz, Azenta Life Sciences (https://www.genewiz.com/), which also provided the shown data analysis.

## STAMP analysis
The Stanford Actionable Mutation Panel of Solid Tumors (STAMP) version 2 sequencing was performed at Stanford University's Anatomic Pathology and Clinical Laboratories.

## RNA-seq
Total RNA was isolated following Trizol reagent RNA extraction protocol for cultured cells. The RNA-seq libraries were constructed by TruSeq Stranded mRNA Library Prep kit (Illumina). All the libraries were sequenced to saturation on Illumina Hiseq2000 or NextSeq sequencers.

## RNA-seq analysis
Fastq files were aligned to hg38 using TopHat 2.1.1 with parameters -p 10 --library-type fr-firststrand -r 100 --mate-std-dev 100. Aligned reads were processed to remove PCR duplicates using Samtools 1.8. Raw counts and RPKM values were calculated using HOMER analyzeRepeats.pl. To test for differential expression, raw reads were compared using DESEQ2, and filtered based on an adjusted $P$ value of <0.05 and twofold change. See Reporting Summary for details.

## qRT-PCR
Total RNA was isolated following Trizol reagent RNA extraction protocol for cultured cells. qRT-PCR was performed as described by TaqMan™ RNA-to-$C_T$™ 1-step kit (Applied Biosystems). Data was collected using Roche LightCycler 480 Software 1.5.1. Data was analyzed using GraphPad Prism v8.2, v9.3 and v9.5.1.

## scRNA sequencing
Differentiated day 7 or enriched and expanded day 50 iSCs were dissociated with Accutase (Innovative Cell Technologies) up to 30 min, filtered with a 40-μm mesh, and washed. Dissociated cells were counted and assayed with trypan blue for live cell counts. Collections with greater than 10% dead cells were processed for dead cell removal using a Dead Cell Removal kit (Miltenyi) following the manufacturer's protocol. A total of 10,000 cells were resuspended in wash buffer at 1000 cells per μL. Library preparation was carried out following the Chromium Single Cell Chromium Next GEM Single Cell 3′ Reagent Kits v3.1 protocol.

## Single-cell RNA-seq analysis
FASTQ files were processed using 10x Genomics Cell Ranger v.3.1.0 and the human genome GRCh38. Cells with UMI counts greater than 500 and with mitochondrial percentage below 20% were included for further analysis. Downstream analyses were performed using Seurat v.4.0.0. 5 iSC samples and a H9 iSC sample were merged into one object and normalized using the default parameters. To compare between each clone, the Seurat object was split by samples and anchors were identified between samples using FindIntegrationAnchors, followed by integration. A total of 2000 highly variable features were identified, objects were scaled to regress out cell cycle stages.

Cells were clustered using 20 dimensions and a resolution of 0.2 to obtain 8 clusters. FindAllMarkers using log(fold change) >0.2 was used for differential expression analysis within individual clusters. For gene scoring, we used the Gibbin-dependent gene list from Collier et al.[28] and the Holoclone geneset from Enzo et al.[40] to run the AddModuleScore function. See Reporting Summary for details.

## Fluorescence-activated cell sorting and flow cytometry
Dissociated cells were washed with FACS buffer (2% BSA Cat# 130-091-376/1 μM ROCK inhibitor/ AutoMACS rinsing solution Cat# 130-091-222). After wash steps, cells were fixed and permeabilized (eBioscience™ Intracellular Fixation & Permeabilization Buffer Set, Cat # 88-8824-00), then stained for antibodies of interest for 30 min at 4 °C (FITC anti-K14 (CBL197 F, Millipore), PerCP anti-K18 (NB120- 7797, Novus), ITGA6 (PE anti-CD49F Cat #555736, BD)), all at 1:100 dilution. Enriched iSCs were analyzed using Streptavidin Alexa Fluor™ 647 conjugate (Life Technologies, Cat # S32357, 1:500 dilution) and FITC-Labeling Reagent (Miltenyi Biotec, Cat # 130-099-136, 1:50 dilution). Composition analysis of enriched iSCs was performed using anti-ITGB4 (BD, Cat # 744150, 1:100 dilution) and anti-CD90 (BD, Cat # 555595, 1:100 dilution). Cells were strained through 35-μm mesh. Flow cytometry data was acquired on a BD LSRII in the Stanford Shared FACS Facility with BD FACSDiva Software (v8.0.1). Ten thousand events were collected. Analysis was performed with FlowJo software (v10.6).

## in vitro skin reconstitution assay
Generation of organotypic epidermis was performed by following the protocol described previously[8] with minor modifications. Derived iSCs and unsorted passaged differentiated cells were expanded in Defined Keratinocyte SFM (Gibco™) until confluent then passaged onto devitalized human dermis at a density of $1 \times 10^6$ cells per 4 cm². The medium was then gradually changed to 7F stratification media for 7 days, after which the dermal sheet was raised to the air-liquid interface. After 2 weeks, the reconstituted epidermis was collected for IF staining.

## Mouse skin engraftment with iPS cell-derived iSCs and Luciferase-RDEB-SCC lines
All animal experiments followed the NIH (National Institutes of Health) *Guide for the Care and Use of Laboratory Animals* under Stanford APLAC (Administrative Panel on Laboratory Animal Care). Xenograft protocol was performed as described previously[8,46]. ITGA6-enriched cells ($1 \times 10^6$) were seeded onto a 1.5-cm² piece of devitalized human dermis (New York Firefighter Skin Bank) and grown in DKSFM (Gibco™) for 10 days, followed by Keratinocyte Growth Medium (Gibco™) for 5 days. Next, the pieces were grafted onto the backs of NOD-NSG mice for 1–9 months. Upon collection, the pieces were embedded in OCT and paraffin for immunofluorescence analysis. In vivo tumorigenicity assays of the reprogrammed and re-differentiated EB-SCC-iSCs expressing the luciferase gene were conducted as previously described[78] and were spiked with normal human keratinocytes (NHKs) to generate the 3D skin constructs and then grafted onto immunodeficient mice. Tumors were formed in mice grafted with RDEB-SCC-iSCs within 4 weeks and detected with the bioluminescence signal intensities using an IVIS optical imaging system. The luminescence intensity of the luciferase assay in regions of transplanted cells from each image was quantified via an automated software process in Living Image software (Caliper, a PerkinElmer company, Columbia University Herbert Irving Comprehensive Cancer Center, New York, NY).

## Alu-qPCR
In grafted mice, detection of emanating cells via Alu sequence was performed on mouse organs, including lymph nodes, liver, lungs, spleen, kidneys, heart, brain, and blood. The organs were resected and transferred to a sterile Petri dish containing cold PBS, washed, finely

minced, and weighed. DNA was extracted from the minced tissue according to the QIAamp DNA mini kit protocol (Qiagen Cat# 51304) with an overnight cell lysis at 56 °C and 10 min lysis for 100 μL blood. Alu-qPCR was performed using 10 ng of extracted DNA in triplicate using amplification protocol by Applied Biosystems (Universal Master Mix II, no UNG: Applied Biosystems Cat# 4440040 and Alu Probe: Thermo Fisher Scientific Cat# 4351372)[48,49].

### Teratoma assay
The teratoma assay was performed as previously described[79]. Briefly, CO1-131 and DEB125-1 iPS cells were cultured as described above, harvested via EDTA-dissociation, and counted in a hemocytometer. $1 \times 10^6$ iPS cells were mixed with Matrigel and injected subcutaneously into immunodeficient mice (https://www.jax.org/strain/017708) that were sacrificed after teratoma formation at the injection site.

### Animal experiments
All animal procedures were approved by the administrative panel on laboratory animal care at Stanford University (APLAC21565). Mice were group-housed (up to 5 mice per cage) on a 12 h/12 h light/dark cycle. Water and standard chow were provided ad libitum. Ambient temperature was 20–22 °C and humidity 20–80%.

### PCR primers
All used PCR primers are listed in Supplementary Table 1 in the Supplementary Information.

### Reporting summary
Further information on research design is available in the Nature Portfolio Reporting Summary linked to this article.

## Data availability
All data from this study will be made available for public access. Patient data will remain de-identified. Bulk RNA-seq, scRNA-seq, next-generation amplicon and whole-genome sequencing data generated in this study have been deposited in the dbGaP database under accession code phs003271.v1. The bulk RNA-seq, scRNA-seq, next-generation amplicon and whole-genome sequencing data are available under restricted access via the dbGaP database, which ensures that only authorized researchers working with appropriate approvals can access datasets derived from human patients. Access can be obtained by following the instructions available at http://www.ncbi.nlm.nih.gov/projects/gap/cgi-bin/study.cgi?study_id=phs003271.v1.p1. Source data are provided with this paper.

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

## Acknowledgements

The authors thank members of the Wernig, Christiano, Steinmetz, Roop and Oro labs for helpful discussions and dedicate this work to Dr. Neehar Bhatia. The authors thank Meredith Weglarz of the Stanford Shared FACS Facility for her expertise in cell purification and flow cytometry. This work was funded by generous grants from the EB Research Partnership (A.E.O., D.R., and A.C.), the California Institute for Regenerative Medicine (TRAN1-10416, DISC2-12590), NIH (ARO73170), CIRM Scholars Program, Department of Defense (W81XWH-18-1-0706), and DEBRA Austria. S.L. and L.M.S. were supported by ERC grant AdG-742804. K.R.R. and T.M.N. were supported by NIH grant R01GM121932. M.M.M. was supported by the German Research Foundation (Deutsche Forschungsgemeinschaft, DFG, MA 8492/1-1). We are grateful to Joel A Jessee, MS (Thermo Fisher Scientific, USA) and Dr. Andreas Reinisch (Medical University of Graz, Austria) for their help with developing and identifying a suitable transfection reagent. Figures 1a, 3f, 5g, and Supplementary Fig. 10a were created with BioRender (BioRender.com; agreement numbers NJ26JJCL70, JV26NTGWB5, XN26JJDR02, and AK26NRGN0P).

## Author contributions

Conceptualization of this work: G.N., J.L.T., S.L., H.H.Z., J.Y.T., L.M.S., M.W., and A.E.O. Production of data: G.N., J.L.T., S.L., K.M., H.H.Z., M.V., S.G., K.M.T., L.L., A.C., G.M.C., K.R.R., and T.M.N. Teratoma assay (Supplementary Fig. 10): G.N., M.M.M., and J.L.T. Tumorigenicity/biodistribution assays (Fig. 5g–i and Supplementary Fig. 10): J.L.T., K.M., J.J.-M., A.R., C.H., Z.G., A.P., and A.C. Cell composition and scRNA-seq characterization: K.M., S.G., and J.L.T. Discussion of gene-editing methodology and derivation of CO1 and CO2 primary patient fibroblasts: G.N., P.S.M., A.B., G.B., and D.R. Development and production of VFS2205: G.G. and T.M.J. Development of reprogramming kit: K.T. Writing, review and editing of manuscript: G.N., J.L.T., S.L., J.Y.T., L.M.S., M.W., and A.E.O.

## Competing interests

K.T. is CEO of iPeace, Inc. M.W. is a scientific advisor for iPeace, Inc. The remaining authors declare no competing interests.

## Additional information

[1]Institute for Stem Cell Biology and Regenerative Medicine, Stanford University, School of Medicine, Stanford, CA, USA. [2]Department of Pathology, Stanford University, School of Medicine, Stanford, CA, USA. [3]Department of Dermatology—Program in Epithelial Biology, Stanford University, School of Medicine, Stanford, CA, USA. [4]Center for Definitive and Curative Medicine, Stanford University, School of Medicine, Stanford, CA, USA. [5]European Molecular Biology Laboratory, Genome Biology Unit, Heidelberg, Germany. [6]Thermo Fisher Scientific, Life Sciences Solutions Group, Cell Biology, Research and Development, Frederick, MD, USA. [7]Department of Dermatology, Columbia University, New York, NY, USA. [8]St. John's Institute of Dermatology, King's College London, London, UK. [9]Department of Genetics, Stanford University, School of Medicine, Stanford, CA, USA. [10]Stanford Genome Technology Center, Stanford University, School of Medicine, Stanford, CA, USA. [11]I Peace Inc., Palo Alto, CA, USA. [12]Department of Dermatology, University of Colorado School of Medicine, Anschutz Medical Campus, Aurora, CO, USA. [13]Department of Chemical and Systems Biology, Stanford University, School of Medicine, Stanford, CA, USA. [14]These authors contributed equally: Gernot Neumayer, Jessica L. Torkelson, Shengdi Li. [15]These authors jointly supervised this work: Marius Wernig, Anthony E. Oro. ✉e-mail: wernig@stanford.edu

