## [Peer Review File · Nature Communications]

A scalable, cGMP-compatible, autologous organotypic cell therapy for Dystrophic Epidermolysis BullosaREVIEWER COMMENTS

Reviewer #1 (Remarks to the Author):

To the authors,

Neumayer and colleagues present in the manuscript „A scalable, GMP-compatible, autologous organotypic cell therapy for Dystrophic Epidermolysis Bullosa“ a GMP-compatible iPS cell-based therapy for the blistering skin disease epidermolysis bullosa. In the presented study the authors showed the possibility to combine the reprogramming and CRISPR-based gene correction step in order to generate COL7A1 corrected clonal iPS cells from primary patient fibroblasts. After conversion into epidermal, dermal and melanocyte progenitors, iPS cell-derived induced skin composite (iSC) grafts onto immune-compromised mice developed into a stable stratified skin. Safety aspects of the application were addressed via targeted and whole genome sequencing as well as toxicologic analyses to detect possible tumor formations in treated mice.

Although the manuscript is well written and the work is of special interest to the scientific community, there are some points, which should be addressed to increase the quality of the manuscript:

- Amplicon NGS analysis of the on-target site would bring more details on the repair outcomes resulting from NHEJ and HDR.
- Figure 1C and D: significances are missing
- Figure 2H: magnification is missing, maybe visualization in colour?
- Figure 3H: please indicate epidermis and dermis in the Figure. Furthermore, the provision of an H&E staining will allow a proper analysis of the polarization and stratification of the epidermis. Staining of other structural proteins within the basement membrane zone (e.g. laminin-332, type XVII collagen, keratin 14 or 5, integrin- $\alpha6\beta4$) is highly recommended.
- Deep off-target analyses at in silico predicted off-target sites highly homologous to the sgRNA binding site(s) should be done on RNP-treated primary patient fibroblasts by amplicon NGS. For the analysis of off-target DNA cleavage in single cell clones sanger sequencing with subsequent TIDE analysis (see Figure 4E) is sufficient.
- Figure 5B: indicate epidermis and dermis. Corresponding H&E stainings are missing. Are the stainings human-specific? Please provide staining of murine tissue as negative control. Please include stainings of an human fibroblast-specific marker such as Vimentin.
- Figure S1: significances are missing
- Figure S3: magnifications are missing
- Figure S5: magnifications are missing

Reviewer #2 (Remarks to the Author):

The field of epidermolysis bullosa therapy is very active, with fundamental and applied advances obtained by several groups, including those of the present authors. They published in 2014 the correction of EB patient keratinocytes by viral editing, after iPSC derivation (Sebastiano 2014), and several groups showed correction and grafting capacity. In the present manuscript, authors present a series of technological advances, notably the successful combination of CRISPR/CAS9-mediated gene editing and reprogramming into a single manufacturing step. Single cell analysis after iPSC differentiation, composite reconstituted skin grafting in mice and comparison of different cell lines grafting potential are shown, and consequences on genomic stability have been particularly well analyzed. A more fundamental finding is the correlation between the presence of specific mesenchymal cells in iSC products and grafting success, but this aspect has not been in depth investigated. The main question is the control used: authors have chosen the human ESC line H9, but a wild type iPSC cell line, subjected to the same type of reprogramming, is required as a correct control. A series of items have to be addressed, notably concerning cell biology, which is the poor part of this manuscript. Notably, the dependence of Gibbin for specific mesenchymal populations is not demonstrated. Ethical information and authorization numbers concerning human biopsies and mouse experimental model are not shown. In summary, this interesting manuscript requires a major revision. Finally, the central question raised by regulatory agencies remains: as the possibility of clinical success has been demonstrated with corrected patient cells grafting, is iPSC cells bioengineering needed for a large application to EB patients, taking into account that such complex bioengineering increases time, cost and genomic stability risk?

Abstract

Line33: The definition of iSC as “iPSC cell-derived organotypic induced skin composite” is somewhat ambiguous in the manuscript, as referring either to cells or to reconstructed skin. This point should be clarified.

Optimization of CRISPR/CAS9-mediated targeting of the COL7A1 locus

Line 131: For the derivation of patient fibroblasts, please add in the materials and methods the anatomical localization of the dermal punch biopsy, as well as the sex for each patient.

Time: authors should be careful with the evaluation of gains in time, and add “at best” at sentences evaluating time required for each step.

Combining iPSC cell-reprogramming and COL7A1 correction in one manufacturing step

L192: Why such differences in number of clones screened for each patient (24-186)? Only one candidate line from 24 clones for DEB125 appears very low.

L218: Why selecting the Colorado mutation? Consequences for the methods, notably the design of the guides?

It would be appreciated to have the content of expected batch release sheet for iPSC and iSC banks. For a clinical application, the list of contaminants (chemicals, viruses...) tested during the production and at release will be necessary.

Scalable and reproducible differentiation of DEB iPSC cells into organotypic skin grafts

L242: Concerning ITGA6 sorting, please add ITGA6 profiling.

L264: Authors have chosen the human ESC line H9, but a wild type iPSC cell line, subjected to the same type of reprogramming, is required as a correct control. Main point.

L267-273: The precise characterization of cell types must be shown, cell biology aspects are poor. C2 : resembling keratinocyte progenitors would be more appropriate than “holoclone keratinocyte stem cells”. Cell phenotype is not sufficiently investigated and data show no functional assay, required for such characterization. Moreover, the percentage of C2 population in Fig3K is not representative of holoclone populations.

C5 and C8: “closely resemble Gibbin-dependent fibroblasts”. Line 46 of the abstract: “transcriptomic revealed prominent Gibbin-dependent signature in dermal fibroblasts”. These two cell populations indeed appear very interesting, notably through a possible correlation with grafting capacity shown in Fig 5. So a better characterization of these cells is lacking, CD90 marker, which is known to be variable among different skin mesenchymal cell populations, is not sufficient. Moreover, authors do not demonstrate that they are Gibbin-dependent. Main points.

Figure 3L and 5C show discrepancies and lack of reproducibility of the proposed protocol on the different cell lines. Authors should discuss this point.

Line 812: “induced keratinocytes were passaged on devitalized dermis at a density of 1×10^6 cells”, please add per what.

In vivo efficacy and favorable safety profile of patient-derived COL7A1-corrected organotypic skin grafts and iPSCs.

L356: Authors should discuss why they have chosen to enrich keratinocytes using ITGA6 sorting instead of rapid collagen adhesion. The first one is best for fundamental studies, but the second one more appropriate for GMP manufacturing.

L358: There is no data on H9 iKC in Fig A table.

L361: Screening Col VII expression at 9 months post-grafting would be necessary.

L369-374: The point that C5 and C8 populations correlate with increased grafting capacity could be a major contribution. However, this point is insufficiently demonstrated in Fig 5 A to D. Notably, CO2-65b number of grafts is too low (4). The tendency should be strengthened by using other cell lines, or the text should be much less affirmative. Main point.

L395: Detecting human Alu sequences in mouse organs is important, but sexual organs must be added, as they are known to be possible tumor targets. Main point.

L396: control with injection of iPSCs alone is missing, in order to check the biodistribution of these cells.

Discussion

L446: This finding is of great interest, and should be proposed as a specification among QC assays.

L458: What could be the type of assay and specification for release, to quantify the different types of cell populations for a future GMP manufacturing?

L464: CD90 marker could be used to validate graft efficiency, but again suppress “Gibbin-dependent” to this sentence. What could be the minimum amount of CD90 to release the batch production?

L510: Authors must add a discussion concerning the implementation of the scaling up in order to produce enough engineered skin to cover all damaged skin surface on an EB patient.

Materials and methods

See above comments on Line 131.

Ethical information and authorization numbers are not included in the manuscript, concerning human biopsies and mouse experimental model. This point is critical.

L528: patient DEB 134 is mentioned in Fig4A.

L551: please add information on iPS culturing, notably passage number, type of selection, and quality controls.

L566: again problems with this cell line, here different culture conditions between H9 and iPSCs.

Figures

L891: this figure is confusing. Data shown are obtained using iSC from H9 or iPS. From 3B to 3H, it would be better to show the same differentiation.

L955: organotypic

L1073: refer the two antibodies used on the dot blot axis.

Reviewer #3 (Remarks to the Author):

This study reports the establishment of a platform of derivation of autologous and genetically corrected organotypic skin grafts for long-lasting treatment of RDEB patient wounds. An important aspect of this platform is the combination of CRISPR/CAS9-mediated gene editing and reprogramming into a single step, reducing the manufacturing step. Another important aspect is the process of inducing corrected iPS cells into skin composite as a whole instead of generating each skin cell types separately. I have only have a few questions below:

The skin composite was shown to include keratinocytes, fibroblasts and melanocytes. However, it is unclear whether the distribution of each cell type in the generated skin composite mimics their distribution normal human skin. Is the density and location of each cell types in the skin composite mimics the human skin? For examples, are melanocytes properly localized to the basal layer of the epidermis in a similar density as normal human skin?

Single cell RNAseq data in Fig. 3L showed that there is strong heterogeneity among individual induced skin composites. Some fibroblast clusters and neuroectoderm clusters are almost completely lacking in certain patient induced skin composites. This raises the concern whether it is better to do the induction of skin composite as a whole (which is difficult to control for a consistent outcome) or induce each cell type separately.

It seems that the authors consider the neuroectoderm cluster to be melanocytes. Are they all expressing mature melanocyte markers or still contain some less differentiated neural-crest like cells?

Itga6+ cells are enriched after day 45 of differentiation. How long does it takes and in what condition is the formation of skin composite afterwards? Method description here could be more detailed.

POINT-BY-POINT RESPONSE TO REVIEWER COMMENTS

Reviewer #1 (Remarks to the Author):

To the authors,

Neumayer and colleagues present in the manuscript „A scalable, GMP-compatible, autologous organotypic cell therapy for Dystrophic Epidermolysis Bullosa“ a GMP-compatible iPS cell-based therapy for the blistering skin disease epidermolysis bullosa. In the presented study the authors showed the possibility to combine the reprogramming and CRISPR-based gene correction step in order to generate COL7A1 corrected clonal iPS cells from primary patient fibroblasts. After conversion into epidermal, dermal and melanocyte progenitors, iPS cell-derived induced skin composite (iSC) grafts onto immune-compromised mice developed into a stable stratified skin. Safety aspects of the application were addressed via targeted and whole genome sequencing as well as toxicologic analyses to detect possible tumor formations in treated mice. Although the manuscript is well written and the work is of special interest to the scientific community, there are some points, which should be addressed to increase the quality of the manuscript:

We thank the reviewer for their support and agree that the work is of special interest to the scientific community. We hope it will provide a roadmap for future iPS cell-derived tissue composite therapies.

- Amplicon NGS analysis of the on-target site would bring more details on the repair outcomes resulting from NHEJ and HDR.

This is a great suggestion! We have conducted this experiment and added the data to the manuscript (Fig S2). The data confirm and extend our previous characterization based on Sanger sequencing. Specifically, the added data confirms sgRNA C4 as the optimal guide for CRISPR-mediated repair of the Colorado mutation and sheds some light on the dynamics of the editing process, i.e. more efficient genomic integration of donor-encoded sequence modifications with more proximity to the CAS9-cut site. The latter has important implications for design of gene editing strategies.

- Figure 1C and D: significances are missing

Initially, Figure 1C and D only contained a n=2 per datapoint. Thus, we did not calculate statistical significances and rather showed the raw data with stdev. As per the reviewer's request, we have now increased sample size to n=6 per data point and calculated statistical significances. As now shown, these experiments proved to be highly reproducible and now provide stronger support for our initial conclusion (i.e. sgRNA C4 being the optimal guide for editing of the Colorado mutation).

- Figure 2H: magnification is missing, maybe visualization in colour?

We have added the missing scale bar but would like to keep the visualization in greyscale as it provides the best contrast in our opinion.

- Figure 3H: please indicate epidermis and dermis in the Figure. Furthermore, the provision of an H&E staining will allow a proper analysis of the polarization and stratification of the epidermis. Staining of other structural proteins within the basement membrane zone (e.g. laminin-332, type XVII collagen, keratin 14 or 5, integrin- $\alpha 6\beta 4$) is highly recommended.

We have now included additional characterization of the regenerated epidermis, including additional K10/K14 immunoreactivity (Figure 3E) and ITGB4 and H&E stainings (in Figure S9).

- Deep off-target analyses at *in silico* predicted off-target sites highly homologous to the sgRNA binding site(s) should be done on RNP-treated primary patient fibroblasts by amplicon NGS. For the analysis of off-target DNA cleavage in single cell clones sanger sequencing with subsequent TIDE analysis (see Figure 4E) is sufficient.

We agree with the reviewer's comment above that amplicon NGS on the on-target locus is a merited experiment adding valuable data to our study. However, we believe that the resolution of amplicon sequencing around *in silico* predicted off-targets is not necessary for this study for the following reasons: Our orthogonal analysis of *in silico* predicted off-targets clearly indicates extremely low to no detectable engagement of the used sgRNA at all predicted off-targets. TIDE analysis from RNP-treated primary patient fibroblasts showed no InDel formation at most off targets when normalized by an internal negative control (Fig 4E,G). The few positive values listed in Fig 4G likely reflect experimental noise that depends on the local quality of the sanger sequencing trace used for TIDE. Specifically, it is expected that background measured at the predicted off target cut site may sometimes be slightly higher than background measured for the internal negative control 100bp upstream of the cut site, resulting in false positives. In agreement with this, a t test comparing all TIDE values measured at predicted off target sites with values measured at internal negative controls was not significant ($p > 0.1$), demonstrating that our approach is not biased for one of the 2 measurement locations. Nonetheless, we can of course not exclude off target mutations entirely. The resolution of NGS may be able to detect super rare off targets and define their nature, but the clonal bottleneck of iPS cell-derivation makes it highly unlikely that such an extremely rare off-target InDel is amplified. In addition, we have built numerous quality controls into our release criteria. This would allow us to exclude iPS cell lines with clonal amplification of off target mutations, should they ever occur. Specifically, all variants detected by NGS (which also focuses on InDels) are screened for homology with the used sgRNA seed sequence. Furthermore, sequencing coverage of off targets is surveyed and shown to detect InDels with single base pair resolution. Finally, we have now also performed targeted Sanger sequencing around all predicted off targets in clonal iPSC lines and found no mutations. In sum, this study focuses on the single manufacturing-step edited/reprogrammed iPS cells and thereof derived therapeutic skin grafts at clinical scale and we demonstrate that our therapeutic product

is free of *in silico* predicted off target mutations that are virtually non-existing in bulk edited fibroblasts and have low probability for clonal amplification. In this context, careful selection of sgRNAs with low to no off-target activity as shown herein is more important than definition of the exact nature of extremely rare off target events.

- Figure 5B: indicate epidermis and dermis. Corresponding H&E stainings are missing. Are the stainings human-specific? Please provide staining of murine tissue as negative control. Please include stainings of an human fibroblast-specific marker such as Vimentin.

We have now included additional characterization of the regenerated epidermis (Figure S9), including additional human-specific markers such as vimentin and H&E stainings. The new data demonstrates that iSC-derived mesoderm contributes to the grafted product.

- Figure S1: significances are missing

Significances have been calculated and added to Figure S1.

- Figure S3: magnifications are missing

We have added the missing scale bars.

- Figure S5: magnifications are missing

We have added the missing scale bars.

Reviewer #2 (Remarks to the Author):

The field of epidermolysis bullosa therapy is very active, with fundamental and applied advances obtained by several groups, including those of the present authors. They published in 2014 the correction of EB patient keratinocytes by viral editing, after iPSC derivation (Sebastiano 2014), and several groups showed correction and grafting capacity. In the present manuscript, authors present a series of technological advances, notably the successful combination of CRISPR/CAS9-mediated gene editing and reprogramming into a single manufacturing step. Single cell analysis after iPSC differentiation, composite reconstituted skin grafting in mice and comparison of different cell lines grafting potential are shown, and consequences on genomic stability have been particularly well analyzed. A more fundamental finding is the correlation between the presence of specific mesenchymal cells in iSC products and grafting success, but this aspect has not been in depth investigated.

We thank the reviewer for highlighting our technological and safety advances as we believe our work will become an important precedence study for similar iPS cell and gene therapies.

1. The main question is the control used: authors have chosen the human ESC line H9, but a wild type iPS cell line, subjected to the same type of reprogramming, is required as a correct control.

We apologize for the confusion, but we have performed our manufacturing protocol using patient fibroblasts from three separate RDEB patients with a particular RDEB mutation, and a fourth primary fibroblast preparation from a patient with a distinct RDEB mutation. We have corrected each of these mutations and generated graftable iSCs providing adequate justification for the process. It was not our intent to include another line as "control", rather we consider the genetically corrected patient iPS cell lines as functionally WT control cells. Given the main problem in the field of line-to-line variability (of "control" cells) we find it much more relevant to demonstrate the efficacy of these 4 corrected EB patient lines than to assess a random ES or iPS cell line. We included the H9 ES cell data for the initial characterization, not to serve as a positive control or a "gold standard", as line-to-line variability excludes such comparisons. However, at the reviewer's request we now include new data for the initial characterization and differentiation from iPS cell lines WTC-11 (<https://hpscereg.eu/cell-line/UCSFi001-A>) and iPSC-DSP (Coriell line AICS-0017) demonstrating that line-to-line variability is prevalent, even among iPS cell lines without any genetic pathogenicity (Fig. S6D). To carry out requested differentiation and characterization of other supposedly wild type ES/iPS cells including toxicology would require an additional 12-24 months of experimentation and, in our opinion, would not add significantly to the already successful use of 5 distinct iPS cell lines from 4 different patients using two different gRNA/RNP/donor combinations.

2. A series of items have to be addressed, notably concerning cell biology, which is the poor part of this manuscript. Notably, the dependence of Gibbin for specific mesenchymal populations is not demonstrated.

We apologize for not making clearer the important connection to our recent publication demonstrating that Gibbin, encoded by the Xia-Gibbs AT-hook DNA-binding-motif-containing 1 (AHDC1) disease gene, interacts with dozens of sequence-specific zinc-finger transcription factors and methyl-CpG-binding proteins to regulate skin mesenchymal differentiation that is critical for the overlying keratinocyte differentiation program (Collier et al. 2022). Notably, Gibbin-mutant human embryonic stem-cell-derived skin organoids lack dermal maturation, resulting in p63-expressing basal cells that exhibit defective keratinocyte stratification. *in vivo* chimeric CRISPR mouse mutants reveal a spectrum of Gibbin-dependent developmental patterning defects affecting craniofacial structure, abdominal wall closure and epidermal stratification that mirror patient phenotypes. These phenotypes are similar to the iSCs from poorly performing RDEB iPS lines CO1-131 and CO1-173. Differential RNA sequencing studies reveal a Gibbin-dependent mesodermal signal matching that of Clusters C5 /C8 of iSCs, as defined in our studies. We have demonstrated this in Figure 5 in the original publication, which is of the entire Gibbin-dependent signature from Collier et al 2022 (please see additional data now in Fig S7). Moreover, RNA sequencing in wild type and Gibbin-mutant iSCs revealed CD90 as a useful marker for Gibbin-dependent mesoderm at D50 in the iSCs. We realize this surface marker is expressed in other

tissues/times during development, but it has been an exceedingly useful potency marker for product clinical development. The precise mechanism how Gibbin acts in CD90+ Gibbin-dependent mesoderm to instruct the overlying basal keratinocytes to promote stratification/graftability is under intense investigation in our lab and will be the subject of upcoming publications.

Ethical information and authorization numbers concerning human biopsies and mouse experimental model are not shown.

These are now included in the manuscript and reporting summary.

In summary, this interesting manuscript requires a major revision. Finally, the central question raised by regulatory agencies remains: as the possibility of clinical success has been demonstrated with corrected patient cells grafting, is iPS cells bioengineering needed for a large application to EB patients, taking into account that such complex bioengineering increases time, cost and genomic stability risk?

We thank the reviewer for this important comment and their overall support of our work. We have had several formal and informal conversations with the FDA (e.g. a recent pre-IND meeting that supports our approach), who have expressed strong encouragement for iPS cell-derived therapies in addition to a series of constructive guidance. In the pre-IND meeting, the FDA indicated that we should include and characterize a Master Cell Bank at the iPS cell stage of our approach, as this will allow repeated production runs for newly occurring wounds and serve as a platform for future therapies that may tackle other tissues affected by the disease (e.g. cornea or esophageal epithelia).

Specifically, the regulatory qualification of this technology will establish a framework on which next-generation iPS cell and gene therapies will capitalize, as iPS cells can be differentiated into other therapeutic products, potentially accelerating clinical translation. For Epidermolysis Bullosa, this may be used to manufacture esophageal or cornea grafts, both tissues severely affected by this disease. As an example, we refer the reviewer to our recently deposited work on ES-derived esophageal mucosa (<https://www.biorxiv.org/content/10.1101/2023.10.24.563664v1>) that forms the basis for using the same patient RDEB iPS clones to manufacture esophageal replacement tissue. iPS cell-mediated engineering of iSCs is also attractive due to the scalability of our approach that could provide virtually unlimited replacement-grafts should the need arise. Furthermore, the high efficiency of iPS cell derivation may overcome problems associated with paucity of graftable patient-derived keratinocytes or holoclones, which may exclude many patients from treatment with corrected patient cells. Finally, as the reviewer has highlighted above, our study has analyzed consequences on genomic stability particularly well and we believe that our manufacturing pipeline will be able to

reproducibly provide a genetically safe product, whereas retroviral safety concerns associated with stochastic insertional mutagenesis remain and might materialize as a possible batch effect.

Abstract

Line33: The definition of iSC as “iPS cell-derived organotypic induced skin composite” is somewhat ambiguous in the manuscript, as referring either to cells or to reconstructed skin. This point should be clarified.

Thanks for this comment. We have clarified this term now better in the manuscript. The proposed drug substance is the iSC that is composed of three lineages. The therapy is the drug substance grafted onto model systems. Application to patient wounds will be developed in the future.

Optimization of CRISPR/CAS9-mediated targeting of the COL7A1 locus
Line 131: For the derivation of patient fibroblasts, please add in the materials and methods the anatomical localization of the dermal punch biopsy, as well as the sex for each patient. Time: authors should be careful with the evaluation of gains in time, and add “at best” at sentences evaluating time required for each step.

The anatomical localizations of dermal punch biopsies are as follows: patient DEB125 – left thigh; patient DEB135 – right hip. Lines CO1 and CO2 were a gift from Dr. Dennis Roop.

As outlined in the reporting summary, “Patient material used in this study has been provided de-identified. Karyotyping revealed that of 4 patients whose cells were used in this study, 2 are genetically male and 2 are genetically female, with no other karyotypic abnormalities. This reflects a balanced distribution of gender for this study.” This information has now also been added to the Material and Methods section.

We have specified timing of the individual manufacturing steps according to our executed production runs, e.g. iPS cell colonies emerged with minimal time-deviations: after 11 days (‘at best’; lines from patient DEB135 and DEB125) and after 14 days (‘at worst’; lines from patients CO1 and CO2).

Combining iPS cell-reprogramming and COL7A1 correction in one manufacturing step
L192: Why such differences in number of clones screened for each patient (24-186)? Only one candidate line from 24 clones for DEB125 appears very low.

On one hand, this reflects the scalability of our approach, theoretically allowing us to screen as many iPSC lines as needed, indicating that our technology would be applicable to loci that may be more difficult to edit. The pragmatic reason is that we first generated 1-step edited/reprogrammed lines from patients CO1, CO2 and DEB135. Due to lacking information on how well isolation of edited iPS cell clones correlates with ddPCR data describing the efficiency of

COL7A1 editing in parental fibroblasts, we picked more clones than necessary. iPS cell line DEB125-1 was established after it was clear that there is a good correlation between editing efficiency in fibroblasts and derivation of 1-step edited/reprogrammed iPS cell lines. Thus, we down-scaled the costly reprogramming-step (only 1 well of a 6 well plate was reprogrammed in this particular experiment) and stopped the screening for clones after a positive hit was obtained.

L218: Why selecting the Colorado mutation? Consequences for the methods, notably the design of the guides?

We picked the Colorado mutation because it is the most frequent mutation in our patient cohort. We did, however, demonstrate that the same approach can be applied to different mutations as shown in the Supplemental Figure S5.

It would be appreciated to have the content of expected batch release sheet for iPSC and iSC banks. For a clinical application, the list of contaminants (chemicals, viruses...) tested during the production and at release will be necessary.

We thank the reviewer for pointing out the need for batch release records for all clinical manufacturing steps. As we look toward IND-enabling studies, we are beginning to address these complex and time-consuming issues, but we feel they are clearly out of the focus of the present manuscript and will be addressed in a future publication with the proof-of-concept clinical trial.

Scalable and reproducible differentiation of DEB iPS cells into organotypic skin grafts
L242: Concerning ITGA6 sorting, please add ITGA6 profiling.

ITGA6 'profiling' is shown in Fig . 3C-E.

L264: Authors have chosen the human ESC line H9, but a wild type iPS cell line, subjected to the same type of reprogrammation, is required as a correct control. Main point.

Please see response to Main Point #1 above.

L267-273: The precise characterization of cell types must be shown, cell biology aspects are poor. C2 : resembling keratinocyte progenitors would be more appropriate than "holoclone keratinocyte stem cells". Cell phenotype is not sufficiently investigated and data show no functional assay, required for such characterization. **Moreover, the percentage of C2 population in Fig3K is not representative of holoclone populations.**

We appreciate the reviewer's comment that the holoclone concentration in keratinocyte co-cultures with J2 mouse feeders is quite low. However, a great advantage of iPS cell-derived skin/iSCs is that we can generate C2 holoclone-like cells in large quantities because we are not limited by mature somatic tissue constraints. This should allow large scale tissue manufacturing approaches for clinical use, a previous hurdle. To further elaborate the similarity of this cluster with physiologically occurring holoclones we now include additional markers from the epidermal, and also dermal and melanocyte, populations in Fig. S7 (also see supplementary Table 1 for used gene signuters).

C5 and C8: "closely resemble Gibbin-dependent fibroblasts". Line 46 of the abstract: "transcriptomic revealed prominent Gibbin-dependent signature in dermal fibroblasts". These two cell populations indeed appear very interesting, notably through a possible correlation with grafting capacity shown in Fig 5. So a better characterization of these cells is lacking, CD90 marker, which is known to be variable among different skin mesenchymal cell populations, is not sufficient. Moreover, authors do not demonstrate that they are Gibbin-dependent. Main points.

We thank the reviewer for this suggestion and refer to our response to Main Point #3 above. In addition, we have improved our product characterization in several ways:

1. Gibbin is now included as a score in the scRNA-seq analysis in Fig. S7.
2. We broadened our product characterization from "CD90 pos" to "CD90 pos/ITGB4 neg" which more closely represents the C5/C8 clusters. This signature also correlates with improved graftability (Fig. 5D and S7).
3. We provide additional data that human vimentin positive mesoderm cells from the product contribute to the murine grafts (Fig S9).

Figure 3L and 5C show discrepancies and lack of reproducibility of the proposed protocol on the different cell lines. Authors should discuss this point.

This is indeed an extremely important point and motivates a deeper mechanistic investigation of epigenetic and genetic effects on manufacturing trajectories, the focus of our work. We have added more discussion on the observed line-to-line and patient-to-patient iPS cell-derived iSC product composition variation.

Line 812: "induced keratinocytes were passaged on devitalized dermis at a density of 1x10⁶ cells", please add per what.

We have added the information in the text.

In vivo efficacy and favorable safety profile of patient-derived COL7A1-corrected organotypic skin grafts and iPSCs.

L356: Authors should discuss why they have chosen to enrich keratinocytes using ITGA6 sorting instead of rapid collagen adhesion. The first one is best for fundamental studies, but the second one more appropriate for GMP manufacturing.

We included ITGA6/B4 as a timing marker for enrichment of mature basal keratinocytes. We used the flow marker as it allows greater quantitation of the manufacturing pipeline compared to rapid adhesion. However, future studies will examine correlative processes including adhesion.

L358: There is no data on H9 iKC in Fig A table.

Please see comment above about “wild type controls”. We used H9 ES cells only to develop the manufacturing platform. As discussed above, we then used RDEB patient iPS cell clones to generate DEBCT product substance and performed the needed animal efficacy and toxicology studies.

L361: Screening Col VII expression at 9 months post-grafting would be necessary.

We regret that this request is technically not possible because the main graft site is closed by this point and the human graft gone. We performed 1, 3, 6, and 9 month scouting toxicology studies using Alu-qPCR to help inform us of rational time points for IND-enabling toxicology, not efficacy studies.

L369-374: The point that C5 and C8 populations correlate with increased grafting capacity could be a major contribution. However, this point is insufficiently demonstrated in Fig 5 A to D. Notably, CO2-65b number of grafts is too low (4). The tendency should be strengthened by using other cell lines, or the text should be much less affirmative. Main point.

We chose 4 grafts because they had high efficacy and were fairly uniform. Since additional grafts will require about another year of work, we have toned down our claims in the text and now highlight that the presence of C5/C8 is only a correlation without known mechanistic implication (although we do not exclude the latter; see also Fig S9). Thank you for pointing out this important issue.

L395: Detecting human Alu sequences in mouse organs is important, but sexual organs must be added, as they are known to be possible tumor targets. Main point.

Regarding the previous grafts, our studies were preliminary toxicology studies that examined the 7 major organs typically involved with emanating cells. These are not definitive IND enabling studies as we have pointed out in comments above. We thank the reviewer for their insightful and helpful comment that we will include when performing IND-enabling studies. However, addition of this extremely time-consuming and expensive additional piece of data will not change the main conclusion that in preliminary studies we did not detect emanation of human cells.

However, we have now performed additional experiments and generated teratomas as per the reviewer's request below. Following subcutaneous injection of RDEB iPS cells, teratomas formed as expected and we have harvested organs and queried for human DNA, including in sexual organs. These results show that no human cells were detected in organs in this assay (Fig. S10).

L396: control with injection of iPSCs alone is missing, in order to check the biodistribution of these cells.

We consider the risk of formation of local or even metastasizing teratomas extremely low given that undifferentiated iPS cells (i) do not survive in Ca²⁺-free, keratinocyte media, (ii) cannot attach and be incorporated into the epithelial sheets that we graft, and (iii) consequently were not detectable in our final cell product (Fig. 5). Nevertheless, we agree that tumor risk is the most important safety concern of this cell product, and the more experimental evidence can be provided the better. We have extensively tested experimentally in over 50 grafts that no local teratomas are formed after transplantation of our cell therapy product. The question at hand thus is whether a local teratoma - if ever generated - would spread into other organs and we assume that this is what the reviewer refers to.

Even if we think this possibility is exceedingly unlikely, we have thought of ways to address this point experimentally. Given that apparently no teratomas are ever formed from our grafts and that it is impossible to incorporate undifferentiated iPS cells into the sheets to be grafted, we decided to generate a local skin teratoma by injecting undifferentiated iPS cells subcutaneously (as this site is anatomically the closest to the proposed DEBCT target graft site). As now shown in supplementary Fig S10F, our specific Alu qPCR confirmed the origin of the teratoma to be human but failed to detect any human DNA in all organs tested (incl. reproductive organs). This suggests that in the highly unlikely event of a DEBCT-induced teratoma, this neoplasia would stay local and could thus be easily eradicated by surgery.

Discussion

L446: This finding is of great interest, and should be proposed as a specification among QC assays. *We suggest to carefully characterize the other allele and target the downstream of the two compound heterozygous mutations to minimize potential undesired consequences of InDel mutations*

We thank the reviewer for their support of our findings and agree. In fact, our data contrasts with reports of significantly higher bi-allelic *COL7A1* editing events in primary RDEB keratinocytes and iPS cells (e.g. PMID: 33609734 and PMID: 31818947). While we cannot exclude that primary keratinocytes and iPS cells may exhibit more favorable NHEJ to ssODN-integration ratios, we note that reported claims of up to 100% bi-allelic correction are based on genotyping by non-quantitative PCR. We urge to carefully characterize the nature of all alleles during quality control, since it is conceivable that undetected large CAS9-induced InDels, which have also been reported by others (e.g. PMIDs: 34365511, 30010673, 33626327, 30937179, 33125898, 30850590, 30089922), may compromise therapeutic safety. We have adjusted the text pertaining to this important point.

L458: What could be the type of assay and specification for release, to quantify the different types of cell populations for a future GMP manufacturing?

We thank the reviewer for bringing up the future need for specification release of the different cell types. While a detailed release plan is out of the scope of this study, we have focused on quantitative single cell assays such as flow cytometry using markers based on single cell transcriptomics.

L464: CD90 marker could be used to validate graft efficiency, but again suppress “Gibbin-dependent” to this sentence. What could be the minimum amount of CD90 to release the batch production?

We show in Collier et al that CD90 is an important Gibbin-dependent transcript (Collier et al 2022) and now show it is highly expressed in C5/C8 clusters (Fig. S7). Based on the current data available we suggest a proposed guide for optimal graftability to be >2% CD90+ cells in the product.

L510: Authors must add a discussion concerning the implementation of the scaling up in order to produce enough engineered skin to cover all damaged skin surface on an EB patient.

Thank you very much for bringing up this important point, that we believe is a key strength of our approach. An important advantage of our approach is that iPS cells are not subjected to the Hayflick limit, thus providing a source for virtually unlimited production of iSC grafts. In a recent landmark study, 1.5×10^9 patient-derived cells transduced with Moloney virus conferring expression of LAMB3 cDNA were used to graft 0.85m^2 of body surface from a junctional EB patient. Given an average coupling efficiency of $\sim 75\%$, we would need to culture 2×10^9 1-step edited/reprogrammed iPS cells to manufacture 1.5×10^9 iSCs. This can be achieved. We have added this important point to the discussion.

Materials and methods

See above comments on Line 131.

Ethical information and authorization numbers are not included in the manuscript, concerning human biopsies and mouse experimental model. This point is critical.

We thank the reviewer for pointing out this omission, as they are in the reporting summary but not the manuscript. All experimental studies were carried out under approved Stem Cell Regulatory Organization, IRB, and APLAC protocols that included detailed patient consents; now included in the methods.

L528: patient DEB 134 is mentioned in Fig4A.

We apologize for this oversight. DEB134 is the sibling of DEB135, who is featured in the supplemental data section. We initially generated edited/reprogrammed iPSC cells from both patients. Unfortunately, DEB134 succumbed to the disease during this study and we continued to only use iPSC cells from DEB135. We have removed the karyotyping results from DEB134 to avoid confusion.

L551: please add information on iPSC culturing, notably passage number, type of selection, and quality controls.

After picking of 1-step edited/reprogrammed iPSCs (P0), we duplicated the lines into sister plates (P1). P1 iPSCs were screened for *COL7A1* editing via ddPCR and positive hits were passaged one more time (P2) for expansion and biobanking. P2 iPSCs were then thawed, expanded, and subjected to iSCs-differentiation (P4). For karyotyping, NGS analysis, P2 iPSCs were thawed and expanded to at least 30×10^6 cells. Our 1-step editing/reprogramming methodology does not use any pharmacological selection. iPSCs were qualified as described in the manuscript (expression of multiple pluripotency markers, karyotyping, teratoma assay etc). Our empirical observation is that our culture system, i.e. StemFit media from Ajinomoto, does not support continuous passaging and culturing of cells other than iPSCs (e.g. primary fibroblasts). This information has been added to the material and methods.

L566: again problems with this cell line, here different culture conditions between H9 and iPSCs.

We thank the reviewer for pointing out this confusion and have now clarified the culture conditions for H9 and iPSC cells.

Figures

L891: this figure is confusing. Data shown are obtained using iSC from H9 or iPS. From 3B to 3H, it would be better to show the same differentiation.

As above, we thank the reviewer for pointing out this confusion and have now clarified the culture conditions for H9 and iPS cells. We have also separated H9 and iPS cell data in Figure 3 and moved other data to Fig. S6 for clarity.

L955: organotypic

We have corrected the spelling.

L1073: refer the two antibodies used on the dot blot axis.

We have corrected the figure and added used antibodies.

Reviewer #3 (Remarks to the Author):

This study reports the establishment of a platform of derivation of autologous and genetically corrected organotypic skin grafts for long-lasting treatment of RDEB patient wounds. An important aspect of this platform is the combination of CRISPR/CAS9-mediated gene editing and reprogramming into a single step, reducing the manufacturing step. Another important aspect is the process of inducing corrected iPS cells into skin composite as a whole instead of generating each skin cell types separately. I have only have a few questions below:

We thank the reviewer for their strong support and agree the work is of importance to the scientific community. We hope it will provide a roadmap for future iPS cell-derived tissue composite therapies.

The skin composite was shown to include keratinocytes, fibroblasts and melanocytes. However, it is unclear whether the distribution of each cell type in the generated skin composite mimics their distribution normal human skin. Is the density and location of each cell types in the skin composite mimics the human skin?

For examples, are melanocytes properly localized to the basal layer of the epidermis in a similar density as normal human skin?

We thank the reviewer for this interesting question. In fact, as discussed with Reviewer #2 above, the protocol does not necessarily produce stoichiometric amounts of each cell type that reflect

normal human skin. For example, melanocytes are present in a 1:7 ratio with keratinocytes in the skin, and that ratio differs with each of the iPS cell lines we tested. The main point of our study was not to address pigmentation or other biologically relevant mechanisms, but the immediate clinically relevant critical quality attribute of manufacturing, i.e. efficacy of graftability on a murine wound. Future studies focus on the interactions of graftable iSCs with melanocytes, but this will be published elsewhere.

Single cell RNAseq data in Fig. 3L showed that there is strong heterogeneity among individual induced skin composites. Some fibroblast clusters and neuroectoderm clusters are almost completely lacking in certain patient induced skin composites. This raises the concern whether it is better to do the induction of skin composite as a whole (which is difficult to control for a consistent outcome) or induce each cell type separately.

Thank you for this comment. While this may seem a logical approach, we found evidence that the co-development of in particular certain mesodermal cells along with basal keratinocyte progenitor cells is needed for optimal maturation and function of the cells. In fact, in earlier approaches we had focused exactly on generating isolated cell types but have now gravitated on generating the cell composite as a whole. Despite the (still large) line-to-line variability for differentiation this approach has proven much more robust than our previous attempts to generate pure keratinocytes. Moreover, a single differentiation protocol may be the only commercially viable path to develop an iPS cell-based therapy given that the costs of goods are already way beyond any currently approved or clinically tested cell therapy.

It seems that the authors consider the neuroectoderm cluster to be melanocytes. Are they all expressing mature melanocyte markers or still contain some less differentiated neural-crest like cells?

We have performed a more detailed analysis of these clusters and found a spectrum of maturation stages. To illustrate this point better we have included additional neural crest/melanocyte markers in the revised manuscript (Fig S7).

Itga6+ cells are enriched after day 45 of differentiation. How long does it takes and in what condition is the formation of skin composite afterwards? Method description here could be more detailed.

We apologize for the lack of details and have now included the information in the revised methods.

REVIEWERS' COMMENTS

Reviewer #1 (Remarks to the Author):

Neumayer et al. have addressed my concerns in a satisfied way. The manuscript is improved now and certainly of interest for the readers.

Reviewer #2 (Remarks to the Author):

The authors took the time necessary to improve the data and the text, and responded to most questions raised by the reviewers. Consequently, this work can now be accepted for publication.

Reviewer #3 (Remarks to the Author):

The authors have addressed my concerns and I have no more questions.